# Cholesterol-binding motifs in STING that control endoplasmic reticulum retention mediate anti-tumoral activity of cholesterol-lowering compounds

Bao-cun Zhang [1] ✉, Marlene F. Laursen [2], Lili Hu[1], Hossein Hazrati[1,3], Ryo Narita [1], Lea S. Jensen[1], Aida S. Hansen [1], Jinrong Huang [4], Yan Zhang[5], Xiangning Ding[1], Maimaitili Muyesier[1], Emil Nilsson [1], Agnieszka Banasik[2], Christina Zeiler [2], Trine H. Mogensen [1,6], Anders Etzerodt [1], Ralf Agger [2], Mogens Johannsen [3], Emil Kofod-Olsen[2,7], Søren R. Paludan [1,7] ✉ & Martin R. Jakobsen [1,7] ✉

The cGAS-STING pathway plays a crucial role in anti-tumoral responses by activating inflammation and reprogramming the tumour microenvironment. Upon activation, STING traffics from the endoplasmic reticulum (ER) to Golgi, allowing signalling complex assembly and induction of interferon and inflammatory cytokines. Here we report that cGAMP stimulation leads to a transient decline in ER cholesterol levels, mediated by Sterol O-Acyltransferase 1-dependent cholesterol esterification. This facilitates ER membrane curvature and STING trafficking to Golgi. Notably, we identify two cholesterol-binding motifs in STING and confirm their contribution to ER-retention of STING. Consequently, depletion of intracellular cholesterol levels enhances STING pathway activation upon cGAMP stimulation. In a preclinical tumour model, intratumorally administered cholesterol depletion therapy potentiated STING-dependent anti-tumoral responses, which, in combination with anti-PD-1 antibodies, promoted tumour remission. Collectively, we demonstrate that ER cholesterol sets a threshold for STING signalling through cholesterol-binding motifs in STING and we propose that this could be exploited for cancer immunotherapy.

The STING (Stimulator of interferon genes) pathway is involved in multiple cellular processes, including interferon (IFN) responses, inflammation, apoptosis, and autophagy[1,2]. STING is ubiquitously expressed in most tissues and cells, where it is located in the endoplasmic reticulum (ER) in the resting state. Upon binding of dsDNA to cyclic-GMP-AMP synthase (cGAS), the enzyme gets activated and synthesize 2′3′-cGAMP, which acts as a second messenger promoting conformational change in STING and its trafficking from the ER

[1]Department of Biomedicine, Aarhus University, DK-8000 Aarhus C, Denmark. [2]Department of Health Science and Technology, Aalborg University, DK-9220 Aalborg, Denmark. [3]Department of Forensic Medicine, Aarhus University, DK-8200 Aarhus N, Denmark. [4]Department of Biology, University of Copenhagen, DK-2100 Copenhagen Ø, Denmark. [5]Department of Engineering, Aarhus University, DK-8000 Aarhus C, Denmark. [6]Department of Infectious Diseases, Aarhus University Hospital, DK-8200 Aarhus N, Denmark. [7]These authors contributed equally: Emil Kofod-Olsen, Søren R. Paludan, Martin R. Jakobsen. ✉e-mail: baocunzh@biomed.au.dk; srp@biomed.au.dk; mrj@biomed.au.dk

to the endoplasmic reticulum–Golgi intermediate compartment (ERGIC)/Golgi through Coat Protein Complex II (COPII) vehicles. The trafficking process is mediated and regulated by numerous proteins, including STING-ER Exit Protein (STEEP), Stromal interaction molecule 1 (STIM1), Toll interacting protein (TOLLIP), Transmembrane P24 Trafficking Protein 2 (TMED2), Yip1 Domain Family Member 5 (YIPF5), and iRhom2[3,4]. At Golgi, TANK-binding kinase 1 (TBK1) is recruited and phosphorylates STING on position Ser366, leading to downstream activation of the transcription factors interferon regulatory factor 3 (IRF3) and NF-κB[5–7]. This ultimately leads to transcription of cytokine genes, particularly type I IFN and proinflammatory cytokines and chemokines[7,8].

Previous work has demonstrated an important role for STING in initiating anti-tumor immune responses[9,10]. In the tumor micro-environment (TME), cancer cells may accumulate cytosolic DNA, leading to elevated cGAMP production and STING activation[11,12]. In parallel, various immune cells infiltrating the TME take up extracellular DNA fragments from dying cancer cells, leading to cGAS-STING pathway activation and immune inflammation in the tumor[13,14]. The folate transporter Solute Carrier Family 19 Member 1 (SLC19A1) and volume-regulated anion channel complex LLC8A:C/E also facilitate the uptake of extracellular cGAMP. It has been suggested that cGAMP produced by cancer cells is released to the extracellular matrix and then taken up by bystander cells. The effect of this is increased dendritic cell (DC) infiltration and activation in the tumor, CD8 T cell priming, and activation through a STING-dependent activation of conventional DCs; the so-called cDC1[15]. Thus, intrinsic STING activation in the TME can be regulated by the amount of cGAMP produced and secreted. This stimulatory effect of extracellular cGAMP can also be markedly increased by blocking hydrolytic degradation of cGAMP, mediated by ecto-nucleotide pyrophosphatase phosphodiesterase 1 (ENPP1)[16], or by various therapies supporting increased DNA damage, such as radiotherapy, chemotherapy, or PARP (poly (ADP-ribose) polymerase) inhibitors.

Direct application of STING agonists into tumors as anti-tumor therapy has been shown to reduce tumor load in several syngeneic mouse models[10,17,18]. Some studies find that cGAMP treatment promotes CD8+ T cell infiltration, IFN-β expression, and tumor regression in wildtype but not in *Sting1*[gt/gt] mice[17,18]. Combining STING agonists with anti-PD-1 checkpoint blockade therapy markedly enhances the efficacy of both therapies in mice[19–21]. Previous phase-I clinical trials investigating the combination of STING agonists in combination with checkpoint inhibitor therapy have indicated some patients with response, but no clinical effect as monotherapy[22,23]. Whether this is due to a lack of elevated anti-tumoral activated immune cells in the tumor, or lack of STING expression or its activation; or a combination of the two, is currently unresolved. Treatment with cyclic-di-nucleotides is complicated by administration route, degradation by ENPP1, limited cellular uptake, bell-shaped dosage curve, and potentially various cytotoxic side effects, especially if given systemically. Consequently, further research into the mechanism of STING pathway activation is still needed to identify more potent strategies that can support activation of the STING pathway and utilize its full therapeutic potential.

Previous research has shown that STING signaling is linked to cholesterol metabolic pathways and cellular cholesterol content[24]. Specifically, reprogramming of lipid metabolism through SREBP2 knockout leads to a reduction of cholesterol synthesis via the mevalonate pathway and increases type I IFN production through STING activation[24]. Moreover, the SREBP cleavage-activating protein (SCAP) and insulin-induced gene 1 (INSIG1), which play central roles in cholesterol homeostasis, have also been found to mediate STING activation[25,26]. Interestingly, deficiency of intracellular cholesterol transporter 1 (NPC1), a factor important for transferring cholesterol from lysosomes to the ER, has been shown to lead to low levels of cholesterol in the ER and priming of STING activation[27]. Taken together, these studies support that cholesterol plays an essential role in STING activation. However, the mechanism by which cholesterol mediates STING activation remains unknown.

In this study, we report that cGAMP stimulation induces reduction in cholesterol levels in the ER, which temporarily lowers the threshold for releasing STING from the ER. This creates a window for STING activation and occurs in a manner dependent on Sterol O-Acyltransferase 1 (SOAT1). Mechanistically, we show that the cholesterol-mediated retention of STING on the ER membrane involve two cholesterol-binding motifs in the protein. Reduction of ER cholesterol, determined by microscopy analysis, leads to enhanced trafficking of STING and a more rapid activation of the downstream pathway, resulting in augmented DC maturation and macrophage activation. Importantly, the combination of cGAMP and the usage of the cholesterol-depleting agent methyl-β-cyclodextrin (MβCD) greatly enhances cellular responses to STING signaling and reduces tumor growth in a murine carcinoma model.

## Results

### Cholesterol impairs STING activation

To investigate whether cholesterol directly affected STING activation, we first established a system in the THP1 monocyte-like cell line for depletion of intracellular cholesterol using MβCD[28]. Determination of ER cholesterol levels were done semi-quantitatively using both confocal microscopy- and ImageStream-based analysis. These analyses revealed a significant reduction in cholesterol levels within ER following MβCD treatment (Supplementary Fig. 1a, b). Building upon this observation, we next conducted a series of treatments on THP1 cells using increasing doses of MβCD. Following cGAMP stimulation, we observed that phosphorylation of STING, TBK1, and IRF3 were clearly increased in cells treated with MβCD (Fig. 1a). To validate if such effect could be achieved in other settings, we treated primary human monocyte-derived dendritic cells with either MβCD or the cellular cholesterol-binding compound, filipin-III, prior to activation with cGAMP. For both treatments we observed elevated type I IFN levels (Supplementary Fig. 1c). We further confirmed that the biological effect of Filipin-III overlapped with the cholesterol-depleting effect of MβCD by its promotion of both STING and TBK1 phosphorylation after cGAMP stimulation, which was most pronounced at early time points (Supplementary Fig. 1d).

Interestingly, exploration of the early steps of STING activation (e.g. dimerization and oligomerization[29]) demonstrated that levels of STING oligomer were enhanced after MβCD treatment (Fig. 1b and Supplementary Fig. 1e upper image), but no effect was observed on STING dimerization (Fig. 1b and Supplementary Fig. 1e lower image). Next, we explored if activation of STING was affected when we depleted two genes known to be involved in various steps of the cholesterol pathway regulation. Lanosterol Synthase (LSS) is involved in cholesterol biosynthesis[30] whereas the ATP Binding Cassette Subfamily G Member 1 (ABCG1), has a critical role in mediating cholesterol efflux[31]. Specifically, using CRISPR-Cas9 editing we disrupted LLS and ABCG1 in THP1 cells prior to stimulation with cGAMP (Supplementary Fig. 2a). As a control we disrupted the safe-harbor gene adeno-associated integration site 1 (AAVS1) locus on chromosome 19[32,33]. Consistent with our results using MβCD, we observed that the phosphorylation levels of STING, TBK1, and IRF3 were augmented in LSS-deficient cells (Fig. 1c). In contrast, the phosphorylation levels were remarkably reduced in cells deficient of ABCG1 (Fig. 1c), which lead to intracellular accumulation of cholesterol. Furthermore, we observed that MβCD treatment of ABCG1-deficient cells restored the phosphorylation levels of STING, TBK1, and IRF3 (Supplementary Fig. 2b). Finally, we demonstrated that MβCD treatment was able to enhance the protein-protein interaction levels of STING and TBK1, sec24, or STEEP under cGAMP stimulation (Fig. 1d, Supplementary Fig. 2c). In support of stronger

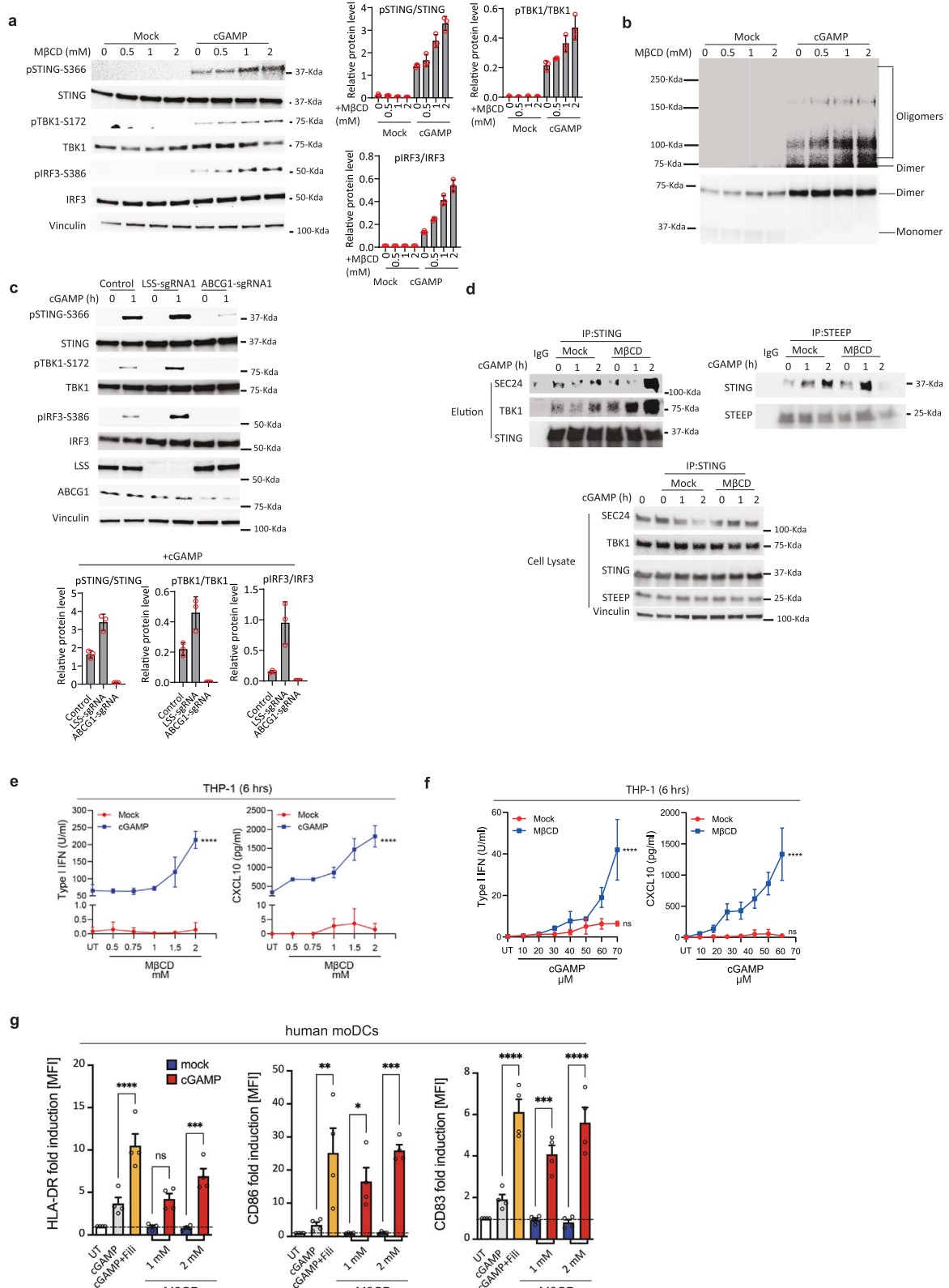

STING pathway activation upon reduction of ER cholesterol levels, we also found that treatment of THP1 cells with MβCD resulted in a significant increase in the secretion of type I IFN and CXCL10 in a dose-dependent manner following treatment with cGAMP (Fig. 1e, f). Importantly, the effects of ~~both~~ filipin-III and MβCD was greatly reduced in THP1 cells deficient in STING (Supplementary Fig. 2d, e), supporting that these two cholesterol-modulating compounds

(though with different biological effects) exerted their effect via STING. To examine whether cholesterol also primed the STING pathway in primary human cells, we next evaluated activation of primary human dendritic cell activation following cGAMP treatment. Importantly, we observed that MβCD augmented cGAMP-induced expression of both HLA-DR, CD83 and CD86 in a dose-dependent manner (Fig. 1g, Supplementary Fig. 3). All together,

**Fig. 1 | Depletion of cholesterol potentiate STING pathway responses.**
**a** Immunoblot analysis of THP1 cells treated with increasing MβCD doses (0–2 mM) before cGAMP (50 μg/ml) stimulation for 1 h. Quantified data presented as mean ± SD from three independent experiments. **b** Immunoblot analysis of STING dimer and oligomer formation in THP1 cells treated with increasing doses of MβCD before stimulation with cGAMP (50 μg/ml, 1 h). (n = 3 biologically independent experiments with similar results). **c** THP1 cells electroporated with Cas9 protein and gene-specific sgRNAs as indicated. Five days post-electroporation, cells stimulated with vehicle or cGAMP (50 μg/ml) for 1 h. Immunoblot analysis performed for indicated protein levels. Quantified data shown as mean ± SD from three independent experiments. **d** Immunoblot analysis of STING, SeC24 or TBK1 co-immunoprecipitated with either endogenous expressed STING or STEEP in THP1 cells. Prior to the immunoprecipitation cells were pre-treated with mock or MβCD (2 mM) and then stimulated with cGAMP (50 μg/ml). IgG from the Sheep/ Rabbit serum was used as negative controls. (n = 2 biologically independent experiments with similar results). **e**, **f** THP1-derived macrophages were treated with cGAMP (5 μM) with varying concentrations of MβCD (**e**) or treated with MβCD (1 mM) and varying amounts of cGAMP (5 μM) (**f**) for a total of 6 h. Cell supernatants were analyzed for IFNα/β by HEK-blue bioassay and CXCL10 expression using ELISA. **g** Flow cytometry analysis of expression levels of maturation markers HLA-DR, CD86, and CD83 in human DC treated with cGAMP (5 μM) with or without filipin-III (1 μg/ml) or methyl-β-cyclodextrin (MβCD) (1 or 2 mM). The experiments for (**a**–**d**) were independently repeated three times with similar results. The representative data from one experiment is shown in the figure. The experiment for (**e**, **f**, **g**) represent one experiment (repeated twice) done in experimental triplicates shown as mean ± SD. Statistical significance in experiment (**e**, **f**, **g**) was assessed using one-way ANOVA with Tukey's multiple comparison correction. (ns not significant; *$p < 0.05$; **$p < 0.01$; ***$p < 0.001$).

these data suggest that cholesterol can play a direct role in regulating the activation of STING signaling.

## cGAMP stimulation reduces ER cholesterol in a SOAT1-dependent manner

Given that cholesterol depletion promotes STING activation, we next explored whether ER cholesterol levels were generally reduced after cGAMP stimulation. Using ImageStream analysis, we found a clear reduction in ER cholesterol levels at early time points (<2h) after cGAMP stimulation that recovered at later time points (>4h) to levels comparable with control cells (Fig. 2a). This was also observed using confocal microscopy, where a significant decrease in ER-cholesterol colocalization was seen at both 1 and 2 h after cGAMP stimulation (Fig. 2b, c). These data prompted us to investigate the mechanism leading to ER cholesterol reduction after cGAMP stimulation. Previous studies have reported several different cholesterol transporters regulating cholesterol levels between ER and other subcellular organelles. Importantly, SOAT1 converts ER cholesterol into cholesterol ester to store in lipid droplets, whereas Sigma-1 Receptor (S1R), facilitates cholesterol export from the ER to mitochondria. Other transporters include Caveolin-1 (CAV1), which delivers cholesterol from the endoplasmic reticulum to the plasma lemma; OSBP transfers cholesterol from the ER to the trans-Golgi network (TGN) or lysosomes; and VAP-A drives OSBP transporting cholesterol from the ER to TGN[34–39]. We applied CRISPR-Cas9 to deplete each of the relevant genes in THP1 cells (Supplementary Fig. 4a). Importantly, we found that SOAT1 deficiency reduced the phosphorylation levels of both STING and TBK1 to the greatest extent compared to the other genes investigated (Fig. 2d, e). Furthermore, our data also suggested that SOAT1 deficiency significantly blocked the reduction of ER cholesterol levels after cGAMP stimulation (Fig. 2f, g). In support of these findings, we observed that knockout of LSS significantly accelerated the reduction of ER cholesterol levels, whereas the knockout of ABCG1 notably hindered the reduction of ER cholesterol levels (Supplementary Fig. 4b). Collectively, these data support that ER cholesterol levels are reduced by SOAT1 at the early stages after cGAMP stimulation to support STING activation prior to the formation of STING oligomerization, phosphorylation, and recruitment of TBK1, which all occur after STING complexes have exited the ER compartment.

## Cholesterol depletion facilitates STING trafficking from ER to Golgi

To investigate the impact of cholesterol on STING trafficking, we next conducted a series of ImageStream analysis to determine the degree of colocalization of STING with the ER marker PDI and the Golgi marker GM130 under MβCD treatment. Interestingly, we observed that cells treated with MβCD had a significant decrease in the colocalization of STING-PDI and an increase in the colocalization of STING-GM130, which was apparent both with and without cGAMP stimulation (Fig. 3a). Next, we introduced the in vitro budding assays previously

reported[4] to test for STING trafficking from the ER. Here, we found that STING was detected to a higher extent in the vesicle fraction when the membrane fraction was from cells pretreated with MβCD (Fig. 3b). These results were further supported by confocal microscopy revealing that STING trafficking from the ER to the Golgi was promoted in cells with cholesterol depletion after cGAMP stimulation (Fig. 3c). Moreover, we observed that the levels of Sec24 foci, which indicate COPII complex formation, were significantly increased after cholesterol depletion. Importantly, the opposite was seen when cholesterol was added to the culture, (Fig. 3d). Taken together, these results support the conclusion that cholesterol depletion enhances COPII complex formation and through this process promote STING trafficking from the ER to the Golgi.

## Cholesterol retains STING at the ER through directly binding to STING

Based on the observations above, we next aimed to investigate the mechanism by which cholesterol mediates the early STING trafficking. Cholesterol can alter the structure of biological membranes in various ways, including changing fluidity, thickness, compressibility, water penetration, and inducing negative curvature of lipid bilayers[40]. Given these factors, we hypothesized that cholesterol depletion could induce alterations in ER morphology by enhancing the curvature of the ER membrane. We explored this by employing the ALPS (Amphipathic Lipid Packing Sensors) motif GFP[133] membrane curvature probe, which has been previously validated for its specific targeting of ER membrane curvature[41]. We observed that cholesterol depletion significantly increased ER membrane positive curvature (Fig. 4a) and also caused the distribution of STING within the cells to increasingly expand along the ER membrane curvature (Supplementary Fig. 5). This has been reported to be crucial for the initiation of STING-COPII complex formation[42]. Furthermore, we observed that cholesterol depletion led to expansion of the tubular ER (Fig. 4b), which is known to be composed of positive curvature in the ER membrane[43]. All together, these data suggest that MβCD treatment and to a lesser extend cGAMP stimulation induced ER cholesterol efflux that facilitate ER membrane curvatures to initiate STING-COPII vehicle formation.

It has been reported that cholesterol interacts with proteins through the Cholesterol recognition/interaction amino acid consensus (CRAC) motif (L/V)-X1 − 5-(Y)-X1 − 5-(K/R) or the inverted CRAC motif, i.e., (K/R)-X1 − 5-(Y/F)-X1 − 5-(L/V)[44]. By screening the amino acid sequence of STING, we found two potential CRAC motifs, $R^{78}xxY^{81}xxxV^{85}$ and $K^{150}xxF^{153}xV^{155}$ (Fig. 4d). The former is located in the transmembrane domain TM2-TM3 linker of STING, and the latter one is located in the connector loop (residues 150–156) linking the connector helix to ligand-binding domain (LBD)α1 of STING (Supplementary Fig. 6a), which allows the proposed 180° rotation of the LBD causing STING activation[45]. To explore this in more details, we conducted a cholesterol pull-down experiment including STING mutants lacking

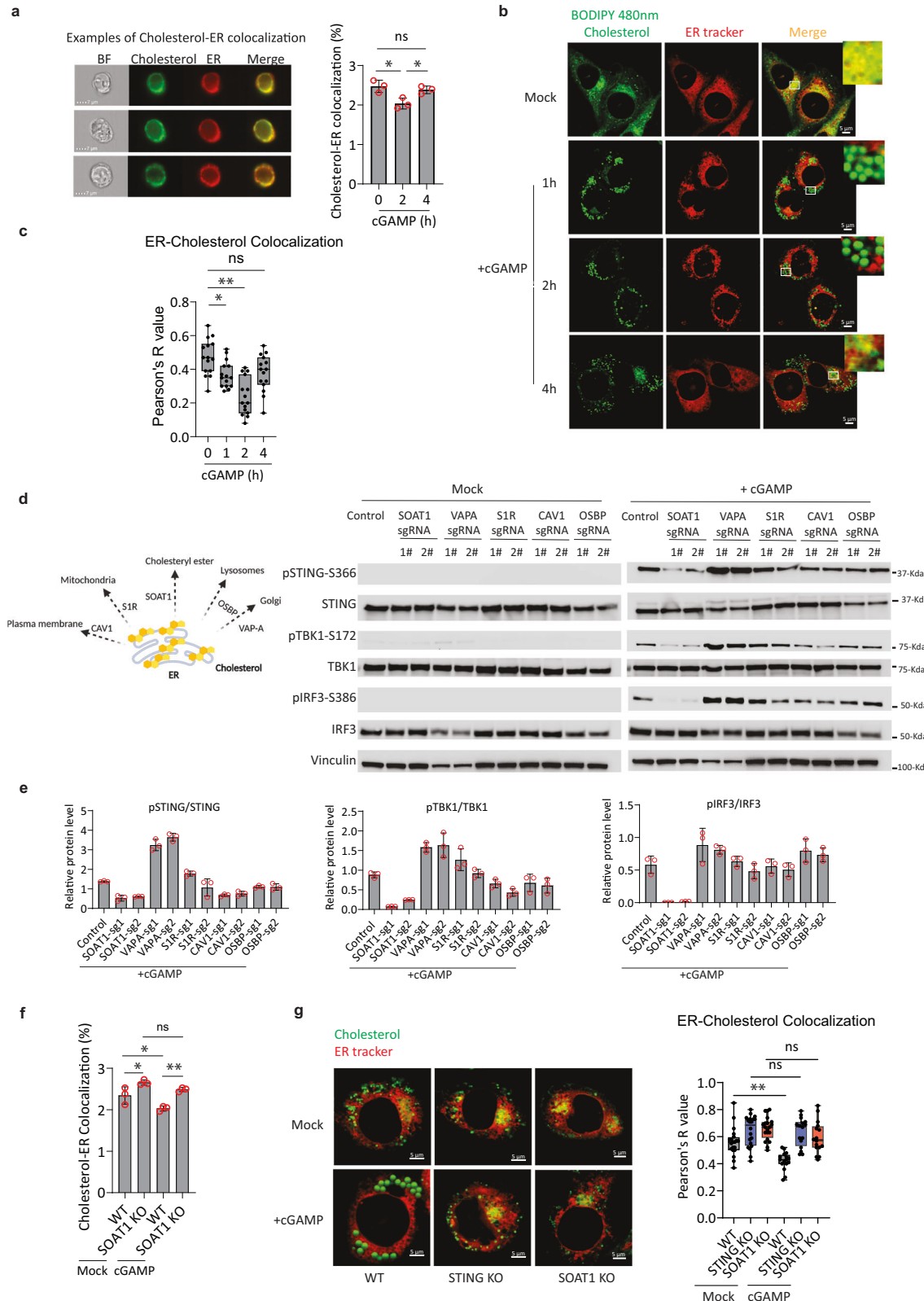

either one of the CRAC motifs (Fig. 4c). Interestingly, while WT STING was efficiently pulled down with cholesterol beads, both STING CRAC mutants exhibited impaired interaction with cholesterol, with the STING K150A-F153A-V155M mutant showing the strongest effect (Fig. 4d, Supplementary Fig. 6b). Consistent with this, mass spectrometry analysis also revealed strong reduction in the levels of cholesterol co-immunoprecipitated with STING, for the mutants, with the

STING K150A-F153A-V155M mutant showing the strongest reduction (Fig. 4e and Supplementary Fig. 7). In addition, all of the above mutants significantly promoted STING-induced ISRE-driven luciferase activity (Fig. 4f), and both STING-Y81A and STING-F153A significantly enhanced STING-ER exit and trafficking to the Golgi (Fig. 4g, h). Together, these data support a mechanism by which cholesterol impairs STING-ER exit through a direct binding to STING CRAC motifs

**Fig. 2 | cGAMP triggers SOAT1-Mediated ER Cholesterol Reduction.**
**a** Colocalization analysis of ER and cholesterol in THP1 cells. Cells were probed with ER-Tracker™ Red and BODIPY-Cholesterol, then either unstimulated or stimulated with cGAMP (50 μg/ml) for 0, 2, or 4 h. Colocalization analyzed using ImageStream. Three cell examples without stimulation demonstrate cholesterol-ER colocalization. **b** Confocal microscopy of THP1 cells pre-stained with Bodipy 480 nm-Cholesterol and ER tracker, then untreated or treated with vehicle or cGAMP (50 μg/ml) for 1, 2, and 4 h. Images acquired using Zeiss LSM 800 confocal microscope, processed with Zen Blue software, and analyzed with ImageJ. **c** 15 images per group in (**b**) quantified using Pearson correlation coefficient tool in ImageJ. Box plot depicts data distribution. **d** THP1 cells electroporated with Cas9 protein and sgRNAs, then stimulated with 50 ug/ml cGAMP for 1 h. Phosphorylation levels of TBK, IRF3, and STING analyzed by immunoblotting. The sgRNA targeting the AAVS1 Safe-Harbor Site was used as control. The experiment was independently repeated three times with similar results. **e** Blot quantification data of (**d**) presented as mean ± SD from three independent experiments. **f** Colocalization analysis of ER and cholesterol in WT or SOAT1 KO THP1 cells using ImageStream. The cells were probed with ER-Tracker™ Red and BODIPY-Cholesterol, then stimulated with vehicle or cGAMP for 2 h. **g** Confocal microscopy of THP1 WT/ STING KO/ SOAT1 KO cells pre-stained with Bodipy 480nm-Cholesterol and ER tracker, then treated with vehicle or cGAMP for 2 h. Pearson correlation coefficient (*r*) quantification performed on 20 vehicle and 15 cGAMP-treated images per group. The experiments for (**a**, **f**) represent one experiment out of three independent experiments, with 10,000 cells per sample for colocalization percentage calculation. Quantitative results are presented as mean ± SD. Statistical analysis conducted with one-way ANOVA. Statistical analysis of (**c**, **g**) performed using one-way ANOVA with Tukey's multiple comparison test. Representative data from one experiment shown (*n* = 2 biologically independent experiments with similar results). (ns not significant; *p < 0.05; **p < 0.01).

and thereby blocks STING trafficking, thus preventing activating of signaling.

## Cholesterol depletion stalls tumor growth in a mouse model in a STING-dependent manner

It has been extensively demonstrated that engagement of the STING pathway can play a significant role in various steps of initiating an antitumor immune response[9,10]. Direct application of STING agonists into tumors has shown to reduce tumor load in several syngeneic mouse models[17,18] dependent on the immunological functions of macrophages and activation of antigen-presenting dendritic cells. Meanwhile, many cancers display altered cholesterol metabolism, and recent studies have demonstrated that manipulating systemic cholesterol metabolism may be useful in improving immunotherapy responses[46]. Therefore, we wanted to explore whether cholesterol depletion could enhance STING-dependent anti-tumor activity in vivo.

We first evaluated if increased STING activation in murine dendritic cells supported better antigen capacity and activation, as we had already shown in human monocyte-derived dendritic cells (see Supplementary Fig. 3b). We found that bone marrow-derived dendritic cells from C57BL/6 mice responded weakly (>10 μM) to free as well as to lipofectamine-delivered cGAMP in terms of MHC-II expression and CD86 activation marker (Supplementary Fig. 8a, b). Interestingly, treatment with low dosage of MβCD significantly increased MHC-II expression more than fourfold and increased CD86 expression twofold, indicating more substantial activation of dendritic cells (Supplementary Fig. 8c). In parallel, we also explored whether the murine colon adenocarcinoma cell line MC38, which contains an active STING pathway, was more responsive following MβCD treatment. Indeed, we observed that MβCD-treated MC38 cells exhibited stronger STING phosphorylation after cGAMP stimulation than control cells (Supplementary Fig. 8d). In all, these data suggested that MβCD treatment may promote a solid anti-tumoral STING-dependent activation in vivo by both priming cancer cells as well as tumor-infiltrating immune cells.

To test this in vivo, we established a solid-flank tumor model using MC38 in C56BL/6 mice and treated them intratumorally with various dosages of MβCD. Interestingly, mice treated with a low dosage of MβCD showed a remarkable delay in tumor growth (Fig. 5a—circle group), whereas higher dosages had no effects (Fig. 5a). We repeated the MβCD monotherapy using a high (30 mg/kg) and low dosage (0.3 mg/kg) and found significant tumor growth delay in the 0.3 mg/kg-treated group (Fig. 5b and Supplementary Fig. 9). This also resulted in an overall better survival profile (Fig. 5c). To demonstrate that this effect was dependent on STING anti-tumoral functions, we next evaluated MβCD treatment in STING-deficient mice (STING-golden ticket). In this model, we did not see any anti-tumor effects, supporting our earlier in vitro data demonstrating that MβCD primes the pathway to respond to low intrinsic STING activating signals (Fig. 5d). These results led us to evaluate the combination of STING agonist with

priming of the tumor microenvironment by MβCD. Here, suboptimal dosage of intratumoral 2′3′-cGAMP treatment showed improved anti-tumor effects when combined with MβCD (Fig. 5e). We next analyzed tumor tissue from mice treated with either cGAMP, MβCD or the combination, to confirm intratumoral STING activation (Fig. 5f). Here, we observed stronger signals for phosphorylated STING as well as TBK1 in tumors treated with the combination, whereas the signals were weaker or absent in the cGAMP-only treated group (Fig. 5f).

Multiple studies have also demonstrated a synergistic effect when targeting both checkpoint receptors and the STING pathway. As our data indicated that monotherapy of MβCD elicited STING-dependent tumor control, we finally evaluated the combination of low dosage of MβCD treatment combined with anti-PD-1 checkpoint blockage. Here, we found that both monotherapy groups had clear tumor growth control during the treatment period; however, the tumors eventually escaped the immune response (Fig. 5g). Nonetheless, in the combination group we did observe a more pronounced immune control (Fig. 5h).

Collectively, these data suggest that cholesterol depletion within the TME can augment STING-dependent anti-tumor activity by increasing STING activation and synergize with checkpoint blockage therapy.

## Human cancers show inverse correlation between cholesterol pathway and ISGs response

To test whether there was connection between cholesterol and STING-dependent ISG response in human cancer, we took advantage of The Cancer Genome Atlas (TCGA) database. First, we selected 15 genes that are typically associated with cholesterol synthesis processes. Then, based on the expression profiles of these genes, all patients from a specific cancer were ranked into four groups. As shown in Fig. 6a, the lowest and highest quarters of gene expression were compared to the expression levels of ISG genes. Using this approach, we found that higher expression of cholesterol synthesis genes was linked to significantly lower expression of ISG genes in multiple cancers (Fig. 6b and S10a–c). Furthermore, given that previous studies have reported higher cholesterol levels potentially facilitate tumor radio-resistance[47–49], we next tested the correlation between cholesterol synthesis genes and ISG genes, using a selective group of radiation-sensitive and -resistant patients from a previous study[50]. We extracted the expression profiles of the SCAP and SREBF genes, which are involved in cholesterol synthesis, and the list of ISG genes (Fig. 6c). From the comparison analysis, we found that the gene expression levels of SCAP and SREBF1 from radiation-resistant patients were significantly higher than those from radiation-sensitive patients (Fig. 6d). In contrast, the ISG gene expression levels from radiation-resistant patients were significantly lower than those from radiation-sensitive patients (Fig. 6e). In conclusion, these results support that high cholesterol synthesis leads to reduced IFN

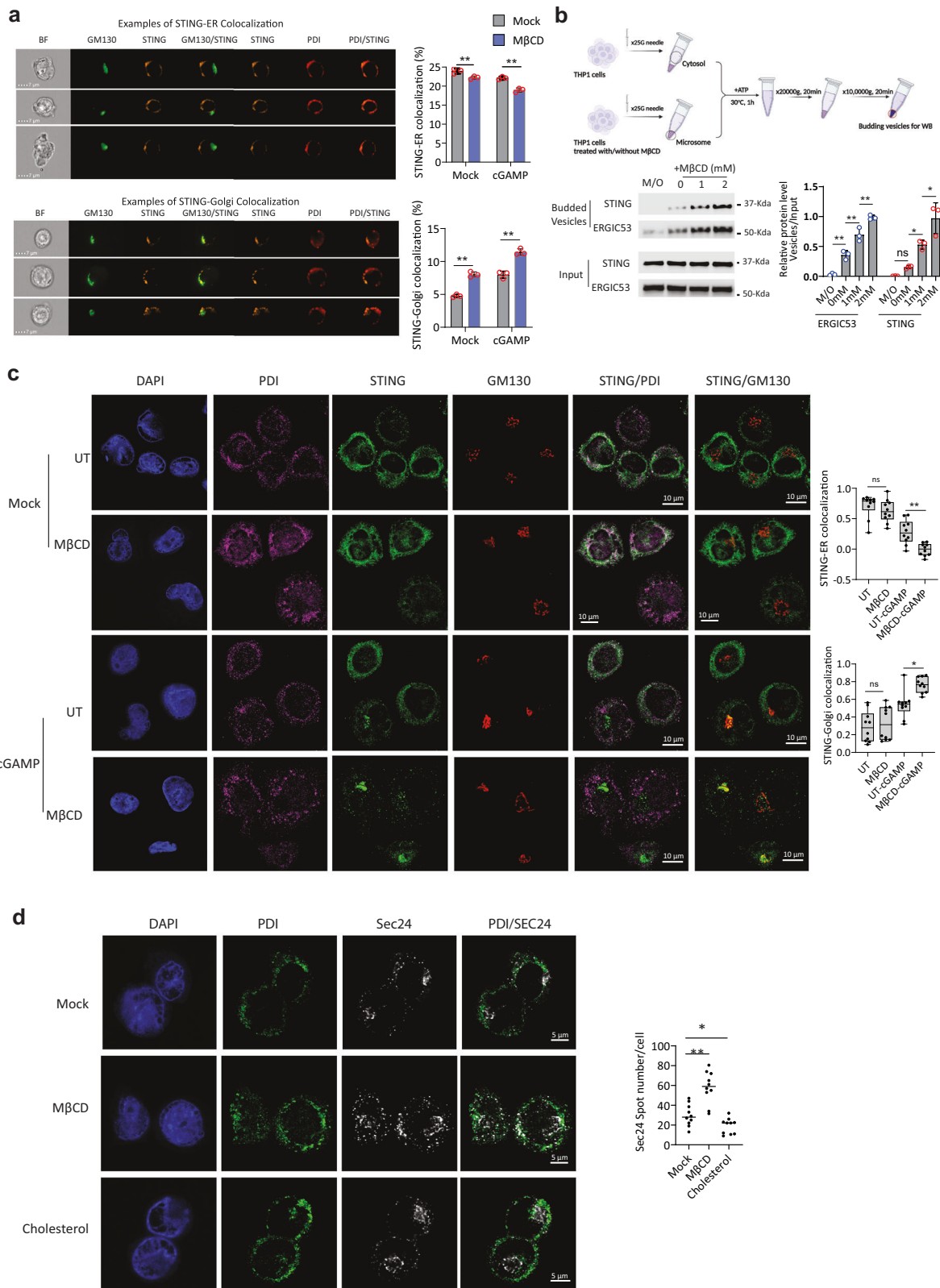

inducible immune responses in the tumor. This could ultimately confer radio-resistance to tumors.

## Discussion

In recent years, the role of lipids in activation and regulation of immune responses beyond the involvement in organelle structures and energy metabolism has been more and more appreciated[51–53].

This includes the involvement in signaling by pattern recognition receptors. Previous studies report that lipid metabolism dysfunction impairs mitochondrial activity leading to mtDNA release and eventually activation of the cGAS-STING pathway[54–56]. Here we report that ER cholesterol interacts directly with STING to limit ER to-Golgi trafficking, thus imposing a threshold on activation of STING. We furthermore show that therapeutic modulation of ER

**Fig. 3 | Depletion of cellular cholesterol facilitates STING trafficking from ER to Golgi. a** HEK293T cells with stable STING expression were treated with or without MβCD (2 mM) and then stimulated with vehicle or cGAMP (50 μg/ml) for 1 h. Fixed with 4% PFA, cells were stained with anti-PDI, anti-STING, and anti-GM130 to analyze STING trafficking from ER to Golgi via image stream. Three cell examples demonstrate the definition of cholesterol-ER colocalization. Quantitative results are presented as mean ± SD. Experiment included three sample replicates under the same conditions, with 10,000 cells per sample collected to calculate colocalization percentage. Statistical analysis performed as two-tailed unpaired *t*-test. **b** In vitro membrane budding assay containing cytosolic fractions from untreated THP1 cells and membrane fraction from microsomes extracted from THP1 cells treated with increasing doses of MβCD (0, 1, 2 mM) for 4 h. Budded materials were analyzed by immunoblotting to assess cholesterol depletion's effect on membrane budding. "Membrane only" (M/O) served as a negative control. Experiment was independently repeated thrice with similar results. Blot quantification data presented as mean ± SD from three independent experiments. Statistical analysis conducted using one-way ANOVA with Tukey's multiple comparisons test. **c** THP1 cells treated with or without MβCD (2 mM) and then stimulated with vehicle or cGAMP (50 μg/ml) for 1 h. Cells probed with anti-PDI (ER marker), anti-GM130 (Golgi marker), and anti-STING, analyzed using Zeiss LSM 800 confocal microscope. Images processed with Zen Blue software 3.8 (Zeiss) and analyzed with ImageJ. Ten images per group quantified through Pearson correlation coefficient (*r*) for STING-ER colocalization and Manders' overlap coefficient for STING-Golgi colocalization in ImageJ. Box plot depicts data distribution. **d** THP1 cells treated with vehicle, MβCD (2 mM, 2 h), Cholesterol (5 mM, overnight), fixed with 4% PFA, stained with anti-Sec24 and anti-PDI. Samples subjected to Zeiss LSM 800 confocal microscope analysis. Ten images per group quantified for Sec24 foci using ImageJ. Statistical analysis of (**c**, **d**) conducted using Statistical analysis of (**c**, **g**) performed using one-way ANOVA with Tukey's multiple comparison test; showing one representative dataset out of two biologically independent experiments, yielding similar results (ns not significant; *$p < 0.05$; **$p < 0.01$).

---

cholesterol levels can be used therapeutically to augment anti-tumor activity of STING agonists.

The role of cholesterol in STING signaling has received significant attention in recent years, and published data have not all been consistent. For instance, it has been reported that removal of cholesterol using MβCD or by disrupting cholesterol flow from lysosomes to ER led to an increase in STING activation[24,27]. However, others suggest that cholesterol contribute to maintaining the lipid composition of the Golgi apparatus, thereby creating favorable conditions for STING activation on the Golgi[57]. Yet another study discovered that cholesterol removal in various in vitro conditions disrupted STING activation[58]. Upon careful examination of the different studies, it appears that reports demonstrating cholesterol to inhibit STING activation have primarily focused on the early stage of STING activation, whereas studies suggesting that cholesterol promotes STING activation have concentrated on the later stage of STING signaling. Thus, we suggest that the timing of STING activation *versus* cholesterol depletion has significant impact on the results, and the different published results are not necessary contradictory. In our study, we depleted cholesterol levels before STING was activated, and thus located in the ER compartment. From our results, we suggest that cholesterol in the ER controls STING activation by preventing STING translocating from the ER to the Golgi at early stage of STING signaling. By contrast, cholesterol in the Golgi membrane may help maintaining the lipid composition of the Golgi, creating a suitable cellular environment for STING phosphorylation at late stage of STING signaling.

A recent study finds that activation of STING upregulates FADS2, which is responsible for the desaturation of polyunsaturated fatty acids (PUFAs). These PUFAs, in turn, inhibit STING activation[59]. In our previous work, we found that phosphatidylinositol 3-phosphate promotes STING trafficking out of the ER, and others have reported an essential role for phosphatidylinositol 4-phosphate in this process[57]. However, direct interaction between lipids and STING activation, and identification of lipid-binding motifs, has not been reported yet. As the STING pathway is involved in infections, autoinflammatory diseases, senescence, and cancer under various conditions, there is a need to expand the knowledge on this pathway. Precise and strict manipulation of STING signaling is essential to balance the induction of stimulus-specific signaling and cell homeostasis in clinical applications. Importantly, here we uncovered that lipid cholesterol in ER binds to STING through a set of distinct cholesterol-binding motifs identified in transmembrane regions of the STING protein (see model proposal in Supplementary Fig. 11). This lipid anchoring inhibits ER membrane curvature, thus impairing STING-ER exit. A previous study had set up a circuit for the lipid metabolic-inflammatory that limits flux through the cholesterol biosynthetic pathway and spontaneously engages a STING-dependent IFN response, while the upregulation of type I IFNs was traced to a decrease in the pool size of synthesized cholesterol[24].

Following this finding, several interesting questions arise: how do type I IFNs decrease cholesterol biosynthesis? How does cholesterol content influence STING? Our finding elucidates the mechanism of how cholesterol negatively influences STING signaling.

One intriguing observation in our study was the transient reduction of ER cholesterol at the early stage of cGAMP stimulation. The efflux of cholesterol from the ER is mediated by SOAT1, which converts cholesterol to inert cholesterol esters, thus establishing a link between cGAMP stimulation and cholesterol level within the ER organelle. The reduction of ER cholesterol levels likely causes both change in the fluidity of the ER membrane, and de-anchoring of STING from the membrane. This in turn facilitates ER membrane curvature and provides a time window for STING trafficking out of the ER. However, the mechanism by which cGAMP stimulation initiates SOAT1-dependent cholesterol esterification still needs to be explored in future studies. In our previous work, we found that the phosphatidylinositol 3-phosphate level on the ER increased after cGAMP stimulation, inducing ER membrane curvature, and initiating the assembly of the STING trafficking complex[42]. Together, these observations indicate that cGAMP binding to STING can cause a complex reprogramming of lipids on the ER, which potentially plays an essential role in the early steps of STING exit from the ER compartment. A notable observation was that the ER cholesterol level returned to a comparable level with the cell resting condition at a later stage of cGAMP treatment. This suggests the existence of a breaking mechanism that recovers the cholesterol level on the ER, potentially hindering over-activation of the STING signaling. One possible explanation for this is that STING trafficking to the lysosome at the late stage of cGAMP stimulation promotes contact between the ER and lysosome, bridged by NPC1. This contact could facilitate cholesterol transfer from the lysosome to the ER and finally resets STING activation[27].

Numerous studies have shown that tumor tissues in breast, thyroid, uterine, ovarian, and renal cancers have higher cholesterol content compared to normal tissues[60]. Cholesterol is essential for the rapid proliferation of cancer cells, and these cells accumulate cholesterol by either up-regulating its biosynthesis or enhancing its uptake[60]. Therefore, reducing cholesterol levels in cancer cells has been proposed as a potential anticancer strategy[61]. Additionally, increasing evidence suggests that targeting the STING pathway can supplement immunotherapy against cancer. STING agonist therapy has been shown to reduce tumor development and increase intratumoral type I IFN production in several mouse tumor models[10,17,18], though often the induction of type I IFN is also associated with increased expression of checkpoint markers[62,63] and thus an elevated immunosuppressive tumor microenvironment. Thus, there is a consensus in the field that anti-tumoral effects of STING therapy will be dependent on combination with checkpoint blockage[19–21,64]. However, STING agonist treatment may also work synergistically with other immunotherapeutic

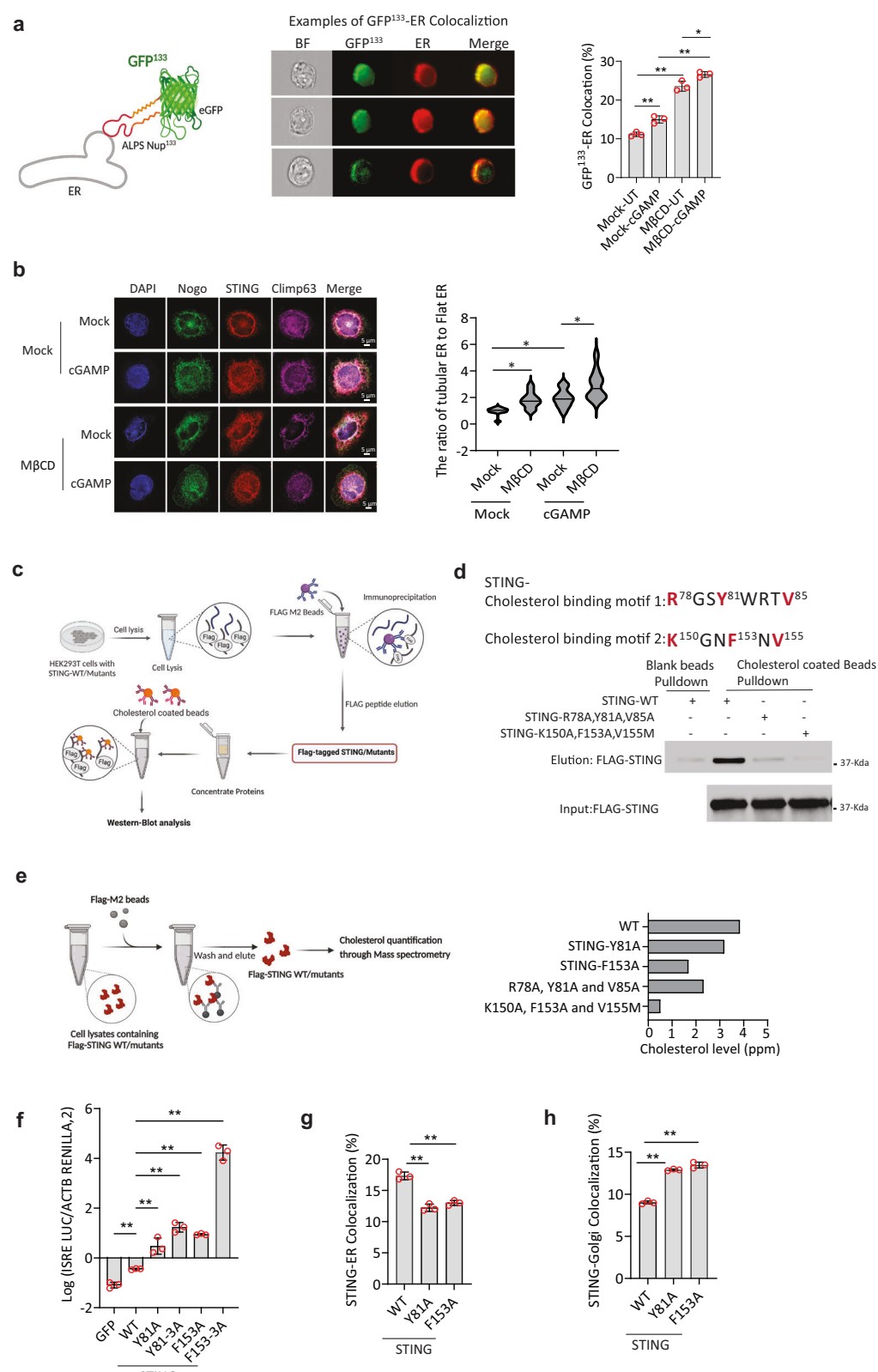

treatments, such as radiotherapy[65], and cancer vaccination[66,67]. Paradoxically, the first two phase-I clinical trials using STING agonists MK-1454 and ADU-S100 in combination with checkpoint inhibitor treatment reported disappointing marginal effects, with no mono-therapy effects[22,23]. This has led to many speculations as to whether STING is a good clinical target. Some suggest that lack of clinical efficacy is due to limited available doses in the tumors. Thus, lowering the threshold of

STING activation may provide a complementary option to enhance the capability of STING agonists in enhancing tumor immunity. Our study provides a new dimension for the combination of STING agonist and interference with cholesterol levels. We found that the application of cholesterol depletion through MβCD in mice not only restricts cancer cell survival and aggressiveness relative to cholesterol levels but also engages the STING agonist-driven anti-tumor response as well as

**Fig. 4 | Cholesterol limits the ER membrane curvature and retains STING at ER through direct interaction. a** GFP-tagged ALPS (GFP[133]) construct transfected into HEK293T cells with stable STING expression. Cells pre-treated with vehicle or MβCD (2 mM) for 2 h, then stimulated with vehicle or cGAMP (50 μg/ml) for 1 h. Fixed cells probed with anti-PDI and analyzed by ImageStream. Three cell examples illustrate cholesterol-ER colocalization. Quantification shown as mean ± SD. The statistical analysis was done using one-way ANOVA with Tukey's multiple comparisons test. **b** THP1 cells pre-treated with MβCD (2 mM) for 2 h, then stimulated with vehicle or cGAMP (50 μg/ml) for 40 min. Fixed cells stained with anti-Clim63, anti-Nogo, and anti-STING for ER morphology analysis. Nogo:Climp63 ratio quantified using ImageJ. Statistical analysis: one-way ANOVA with Tukey's multiple comparison, including a two-sided test. Data represent one experiment out of two independent experiments with similar results. **c** Schematic of pull-down assay using cholesterol-coated beads. **d** The level of STING protein bound to cholesterol was determined by immunoblotting (shown as a representative image out of $n = 3$

experiments). **e** Mass spectrometry analysis of cholesterol binding on STING. Immunoprecipitation with FLAG-bead using HEK293T lysates expressing Flag-tagged STING-WT or mutants. Cholesterol levels quantified ($n = 2$ independent experiments) – see chromatogram in Fig. S7. **f** ISRE reporter gene assay in HEK293T cells with STING stable expression. Luciferase activity measured using Dual Glo kit. Quantitative results are presented as mean ± SD. Data represent one experiment out of three independent experiments. **g**, **h** ImageStream analysis of STING-ER (**g**) and STING-Golgi (**h**) colocalization. The experiments for (**a**, **g**, **h**) are represented from one experiment (out of a total of 3 independent experiments) with 10,000 cells per sample collected to calculate the colocalization percentage. Quantitative results are presented as mean ± SD. The statistical analysis was done with one-way ANOVA. The statistical analysis of (**f**, **g**, **h**) was done using one-way ANOVA with Tukey's multiple comparison correction, and data was illustrated as mean ± SD (ns, not significant; *$p < 0.05$; **$p < 0.01$, ***$p < 0.001$).

checkpoint blockage. Surprisingly, we found that high levels of MβCD was counteracting the anti-tumoral functions compared to low doses MβCD. We speculate this could be due to regulating the balance between elevated STING activation and the negative effects of high cholesterol depletion on immune cell functionalities[68]. Further work will be needed to identify the sweet spot that support an overall anti-tumoral response in the TME.

In conclusion, we propose that ER cholesterol imposes a threshold for STING trafficking and activation through direct binding to CRAC motifs in the STING protein primary sequence. The ER cholesterol levels are transiently reduced upon STING agonist treatment, thus creating a window with lower threshold for STING trafficking and thus activation. We demonstrate that this can be harnessed therapeutically, by lowering cellular cholesterol levels and represent an alternative approach to improve the effect of cancer immunotherapy.

## Methods

### Ethical statement

Buffy coats from healthy donors were anonymously obtained from the blood bank at Aarhus University Hospital, Skejby, or Aalborg University Hospital, Denmark. The animal studies were conducted in accordance with The Danish Animal Ethics Council, license number: 2017-15-0201-01253.

### Mice and in vivo experiments

Six-weeks-old female C57BL/6 mice were purchased from Janvier. Upon arrival, animals were allowed to acclimatize for two weeks. All animals were housed under specific pathogen-free conditions, with SPF status checked regularly. Light cycle was 12:12 h light/dark with light from 6 am till 6 pm; temperature between 20 and 24 degrees Celsius and humidity at 55% which were monitored by air handling unit Scanclime (Scanbur, Denmark). Animal cages were housed in IVC racks from the GreenLine series (Techiplast) using GM500 or GR900, including Edstrom automatic watering system (Avidity Science). Water applied to animals was reverse osmosis water with added chlorine. The level of active/free chlorine was between 0.8 and 2.0 ppm. Regular chow was from Altromin; product #1324 in 10 mm pellets. All chow was autoclaved locally, and thus feed was supplied as 'enriched', i.e. with extra nutrients in relation to the animals' actual needs. All experimental protocols were approved by the Danish Animal Experimentation Council and performed according to national guidelines. Approximately 8-weeks-old female C57BL/6 mice were inoculated subcutaneously with $1 \times 10^6$ MC38 cells/ml in PBS in the right flank. When tumors had reached an average volume of 85 mm³, mice were randomized into groups, dependent on the experimental setup. Treatment of MβCD as well as STING agonist 2'3'-cGAMP (invivogen Vaccigrade) was done by intratumorally injections (i.t.). Treatment with anti-PD-1 antibody (RMP1-14) or PBS as control was done

intraperitoneally (i.p.). Tumors were measured with a caliper every third day, and tumor volumes were calculated as π/6 × length × width². Mice were euthanized by cervical dislocation when tumors exceeded 900 mm³ volume or they developed excessive wounds (combination therapy experiment).

### Cell cultures

MC38 (ATCC cat no. T8291) and human monocytic cell line THP1 (ATCC cat no. TIB-202) WT and STING KO (51) cells were cultured in RPMI 1640 medium (Thermo Fisher) supplemented with heat-inactivated 10% FBS (Sigma–Aldrich), 2 mM L-Glutamine and 50 μg/ml streptomycin (Sigma–Aldrich), 50 U/ml penicillin (Sigma–Aldrich) (hereafter referred to as complete RPMI). THP1-derived macrophages were generated by culture in RPMI medium containing 100 nM Phorbol 12-myristate 13-acetate (PMA) (Sigma–Aldrich) for 24 h, followed by 24 h incubation in PMA-free medium. HaCaT (ATCC cat no. PCS-200-011) and HEK293T (ATCC cat no. CRL-12585) were cultured in DMEM (high glucose) medium (Thermo Fisher) supplemented with heat-inactivated 10% FBS (Sigma–Aldrich), 50 μg/ml streptomycin (Sigma–Aldrich), and 50 U/ml penicillin (Sigma–Aldrich). Human monocyte-derived dendritic cells (moDCs) were generated from whole blood, collected from healthy volunteer donors in sodium-heparin tubes (Sarstedt). Peripheral blood mononuclear cells (PBMCs) were isolated by density centrifugation using Lymphoprep (Axis-Shield). The Pan Monocyte Isolation Kit, Human (Miltenyi Biotech) was used to isolate untouched monocytes from PBMCs, which were differentiated by culturing in complete RPMI supplemented with 400 U/ml IL-4 and 1000 U/ml GM-CSF for 6 days, with fresh medium added to the cells on day 3.

Bone marrow-derived dendritic cells were generated from bone marrow isolated from C57BL/6 female mice (Janvier Laboratories). Mice were sacrificed by cervical dislocation and washed in ethanol. The femur and tibia bones were removed from both legs and tissue was cleaned off. Bones were cut at distal ends and the bone marrow was flushed out with PBS. Bone marrow cells were washed once in PBS, transferred to culture medium, and seeded in 10-cm petri dishes. BMDCs were differentiated by culturing for 10 days in complete RPMI supplemented with 400 U/ml IL-4 (Miltenyi Biotech), 1000 U/ml GM-CSF (Miltenyi Biotech), and 5 μM β-mercaptoethanol (Sigma–Aldrich). Fresh medium was added on days 3, 5, and 7. Bone marrow-derived macrophages were generated from bone marrow isolated from C57BL/6 mice. Cells were resuspended in RPMI and run through a 70-μm cell strainer, followed by one wash before seeding cells in complete RPMI supplemented with 20% L929 supernatant (containing M-CSF). Cells were kept in culture for 6 days. Medium was changed on day 4 to RPMI supplemented with 40% L929 supernatant. On day 6, cells were harvested using Accutase Solution (Sigma–Aldrich) and reseeded in well plates. All cells were kept in a humidified incubator at 37 °C, 5% $CO_2$.

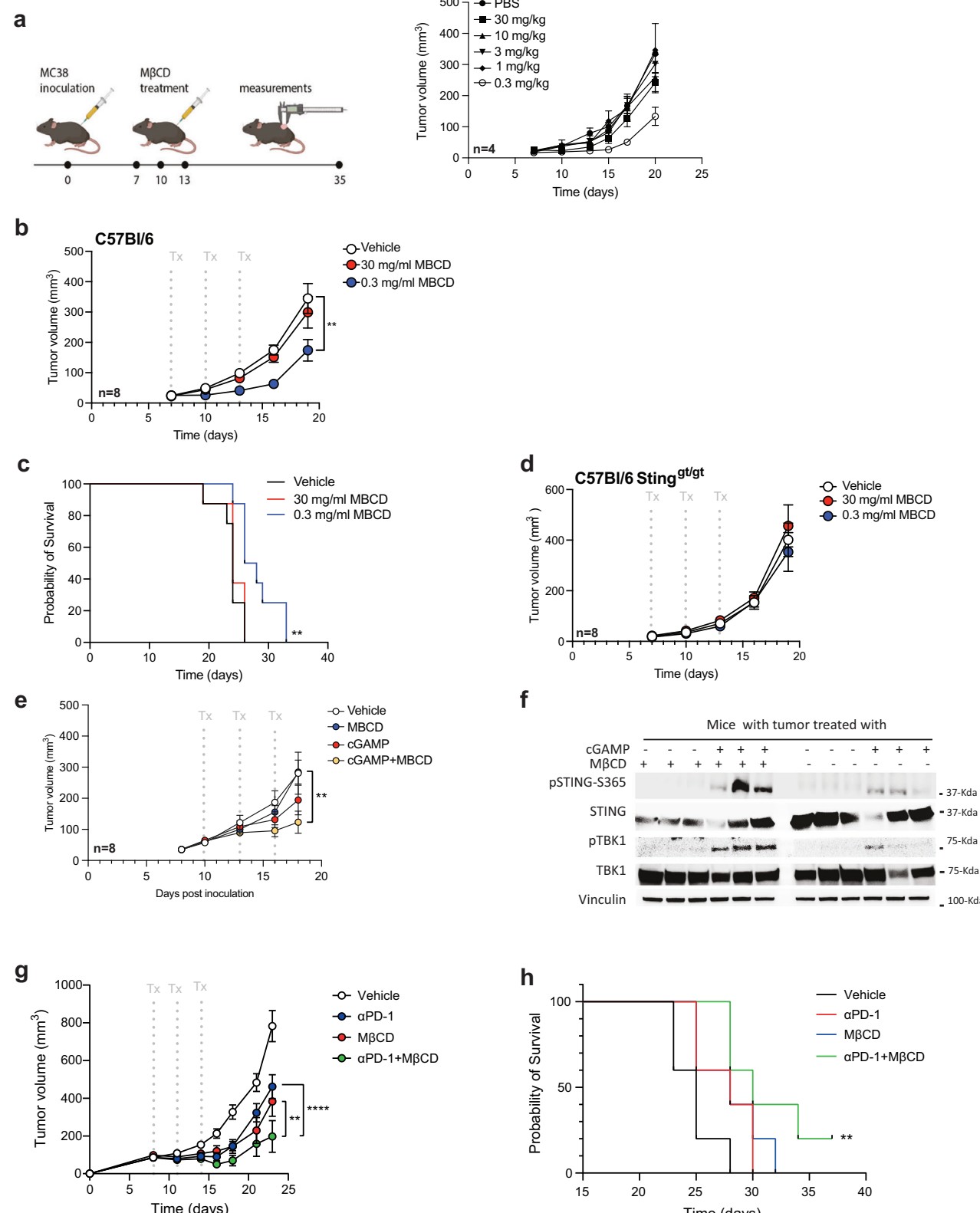

## Generation of genome-edited cell lines

For the gene knockout in THP1 cells using CRISPR/Cas9 technology, chemically modified synthetic sgRNAs were obtained from Synthego. The specific sequences of the utilized sgRNAs are detailed in Supplementary Table 1. To introduce the sgRNAs into the cells, electroporation was carried out with the 4D-Nucleofector device from Lonza, utilizing a 20 µL format Nucleocuvette Strip. For each electroporation reaction, $1.5 \times 10^5$ THP1 cells were utilized, along with 1 µg of the sgRNA and 0.6 µL of Alt-R™ S.p. Cas9 Nuclease V3 (10 µg/µL, Integrated DNA Technologies). The electroporation buffer Opti-MEM (Gibco) and the program P3-CM138 were employed in this process. The assessment of knockout efficiency was performed using the Inference of CRISPR Edits (ICE) method.

**Fig. 5 | Cholesterol depletion through Methyl-β cyclodextrin in vivo stalls tumor growth in a STING-dependent manner. a** C56BL/6 mice ($n = 5$) were inoculated with $1 \times 10^6$ MC38 colon adenocarcinoma cells on right flank and treated with various doses of MβCD by intratumoral injection on day 7, 10, and 13. Tumor growth is shown as mean ± SEM. **b** C56BL/6 mice ($n = 8$) with MC38 tumors were treated with high (30 mg/kg) and low (0.3 mg/kg) dose of MβCD by intratumoral injection on day 7, 10, and 13. Tumor growth is shown as mean ± SEM. The individual spaghetti plots can be seen in Supplementary Fig 9. Statistically significant differences were calculated using a two-way ANOVA with Geisser-Greenhouse correction followed by Holm−Sidak's multiple comparisons test (**$p < 0.01$). **c** Kaplan−Meier plot depicting probability of survival in each of the treatment groups. Statistically significant difference was calculated using Mantel-Cox Log-rank test. (**$p < 0.01$). **d** Tumor volume measurements in C57BL/6 STING-deficient mice ($n = 6$) (the golden ticket strain) following same treatment regime as wildtype mice. Tumor growth is shown as mean ± SEM. **e** C56BL/6 mice ($n = 8$) with MC38

tumors were treated with MβCD (0.3 mg/kg) or 2'3'-cGAMP (1 μg/dosage), or the combination by intratumoral injection on day 10, 13, and 16. Tumor growth is shown as mean ± SEM. Statistically significant differences were calculated using two-way ANOVA with Holm−Sidak's multiple comparisons test (**$p < 0.01$). **f** Immunoblot analysis of STING and TBK1 phosphorylation levels in tumors ($n = 3$ different mouse/group) from mice treated with MβCD (0.3 mg/kg) or 2'3'-cGAMP (1 μg/dosage), or the combination by intratumoral injection. **g** C56BL/6 mice ($n = 5$) with MC38 tumors were treated with three different treatment schemes on day 8, 11, and 14. Treatment consisted of (i) 0.3 mg/kg dose of MβCD delivered by intratumoral injection, (ii) 10 mg/kg anti-PD-1 antibody (RMP1-14) delivered as intraperitoneal injection, or (iii) a combination of MβCD and RMP1-14. Tumor growth is shown as mean ± SEM. Statistically significant differences were calculated using two-way ANOVA with Holm−Sidak's multiple comparisons test (**$p < 0.01$; ****$p < 0.0001$). **h** Kaplan−Meier plot depicting *p*robability of survival. Statistically significant difference was calculated using Mantel-Cox Log-rank test. (**$p < 0.01$).

To perform the SOAT1 knockout in THP1 cells via lentivirus delivery, lentiviral vectors were generated following previously established protocols[69]. HEK293T cells were transiently co-transfected with four plasmids: the relevant lentiviral vector, pMD.2 G, pRSV-Rev, and pMDlg/pRRE-D64V, along with the gRNA-lentiCRISPRv2 targeting SOAT1. Lipofectamine 2000 was employed as the transfection reagent in this process. Following a 48-h incubation period, the supernatant was harvested, and the lentiviral particles were concentrated using the 5xLenti Concentrator (ORIGENE). To establish stable gene knockout cell lines, THP1 cells were transduced with the lentivirus. After 48 h of infection, the cells underwent selection with puromycin at a concentration of 0.5 μg/ml. Then, the cells were assessed for KO efficacy.

To establish HEK293T cells expressing STING-WT/mutants stably, we initiated the process by transiently co-transfecting HEK293T cells with four plasmids: the pertinent lentiviral vector, pMD.2 G, pRSV-Rev, and pMDlg/pRRE-D64V, in conjunction with the lentivector containing the STING-WT/mutants. Following a 48-h incubation period, we collected the supernatant and subsequently concentrated the lentiviral particles using the 5xLenti Concentrator (ORIGENE). Subsequently, a fresh batch of HEK293T cells was transduced with the lentivirus. After a 48-h infection period, the cells were subjected to selection using blasticidin at a concentration of 2 μg/ml.

### Stimulation with drugs
**cGAMP stimulation delivered by lipofectamine.** DCs and THP1 cells were stimulated with 2'3'-cGAMP and methyl-β-cyclodextrin (Sigma−Aldrich) or filipin-III from *Streptomyces filipinensis* (Sigma−Aldrich), for 1 h with 0.05–2 mM methyl-β-cyclodextrin or 100–1000 μg/ml filipin-III before stimulation with 1–50 μM 2'3'-cGAMP delivered by Lipofectamine 3000 (Thermo Scientific).

**cGAMP stimulation added to medium.** THP1 cells were pre-treated for 1 h with 0.05–2 mM methyl-β-cyclodextrin before stimulation with 50 μg/ml 2'3'-cGAMP.

### IFNα/β bioassay and ELISA
HEK-blue cells (InvivoGen) were cultured in DMEM medium, supplemented with 10% FBS, 2 mM L-Glutamine, 50 μg/ml streptomycin, 50 U/ml penicillin, and 100 μg/ml normocin (InvivoGen), 30 μg/ml blasticidin (Thermo Fisher) and 100 μg/ml zeocin (InvivoGen). For the IFNα/β bioassay, samples were added to flat bottom 96-well plates, and 2–500 U/ml IFN-α (Abcam) was included in wells as a positive control and to generate standard curves. Plates were incubated overnight, and QUANTI-Blue Solution (InVivoGen) was prepared according to the manufacturer's instructions and added to the wells. Supernatants from the induced HEK-Blue IFNα/β cells were added to the plates containing QUANTI-Blue Solution and incubated at 37 °C for 5–30 min. OD was measured using a spectrophotometer at 620–655 nm.

Human CXCL10 was measured using the Human CXCL10/IP-10 DuoSet ELISA kit (RnD Sytems) and murine CXCL10 and IFN-β was measured using Mouse CXCL10/IP-10/CRG-2 DuoSet ELISA (RnD Sytems) and Mouse IFN-beta DuoSet ELISA (RnD Sytems), respectively, according to kit protocols.

### Flow cytometry
Cells were stained with Fixable Viability Dye eFluor 780 (Thermo Fisher Scientific) in PBS for 30 min at 4 °C. Cells were stained with antibodies or corresponding isotype controls, for 30 min at 4 °C. Antibodies for flow cytometry analysis of human moDC maturation were mouse IgG1 κ anti-human HLA-DR PE (RnD Systems), mouse IgG1 κ anti-human-CD83-PE-Cy7 (BD Bioscience), mouse IgG1 κ anti-human CD86 BV421 (BD Bioscience), mouse IgG1 κ isotype control PE, PE-Cy7 or BV421 (BD Bioscience). Antibodies used for flow cytometry analysis of murine BMDC maturation were rat IgG2b κ anti-mouse-MHC Class II (I-A/I-E)-PE (Thermo Fisher Scientific), rat IgG2a κ anti-mouse-CD86-eFluor 450 (Thermo Fisher Scientific), rat IgG2b κ isotype control PE (Thermo Fisher Scientific) and rat IgG2a κ isotype control eFluor 450 (Thermo Fisher Scientific). Cells were analyzed using a Cytoflex S2 flow cytometer (Beckman Coulter). Cells were gated using FSC-SSC, and singlets were gated using FSC-A-FSC-H. At least 10,000 events were collected in the live cell gate. Data were analyzed in Kaluza software (Beckman Coulter) and GraphPad Prism 6.

### Western blot
A minimum of 100,000 cells were lyzed in 100 μl ice-cold RIPA Lysis and Extraction Buffer (Thermo Fisher Scientific), supplemented with Complete Protease Inhibitor Cocktail (Sigma−Aldrich), Pierce Protease and Phosphatase Inhibitor (Thermo Fisher Scientific), 10 mM Sodium Fluoride and Benzonase (Sigma−Aldrich).

The tumor tissues were homogenized using steel beads from Qiagen in a Tissuelyser (II) (Qiagen). Subsequently, the homogenized samples were sonicated for 5 min. Following this step, the samples were combined with RIPA buffer (Thermo) with Complete Protease Inhibitor Cocktail (Sigma−Aldrich), PhoSTOP (Sigma), Benzonase (Sigma−Aldrich), and 0.1% SDS. Then, the lysates were subjected to centrifugation at 14,000 g for 10 min at 4 °C. After centrifugation, the supernatant was carefully collected and preserved for use as the lysate. Lysate were denatured in 2× Laemmli buffer (Sigma−Aldrich) with DTT or in 4× XT sample buffer (BioRad) and 20× detergent (BioRad) and heated at 95 °C for 5 min. Protein separation was done by SDS-PAGE gel electrophoresis using Criterion TGX Precast Gels gels (BioRad). The gel was transferred onto 0.2 μm PVDF membrane using a Trans-Blot-Turbo transfer system (BioRad). Membranes were blocked with TBST + 5% skim milk, followed by staining with primary antibodies diluted (1:1000) in TBST + 5% skim milk. Membranes washed in TBST before staining with secondary antibodies diluted (1:7500) in TBST + 5% skim milk. Membranes were exposed with Super Signal West Femto

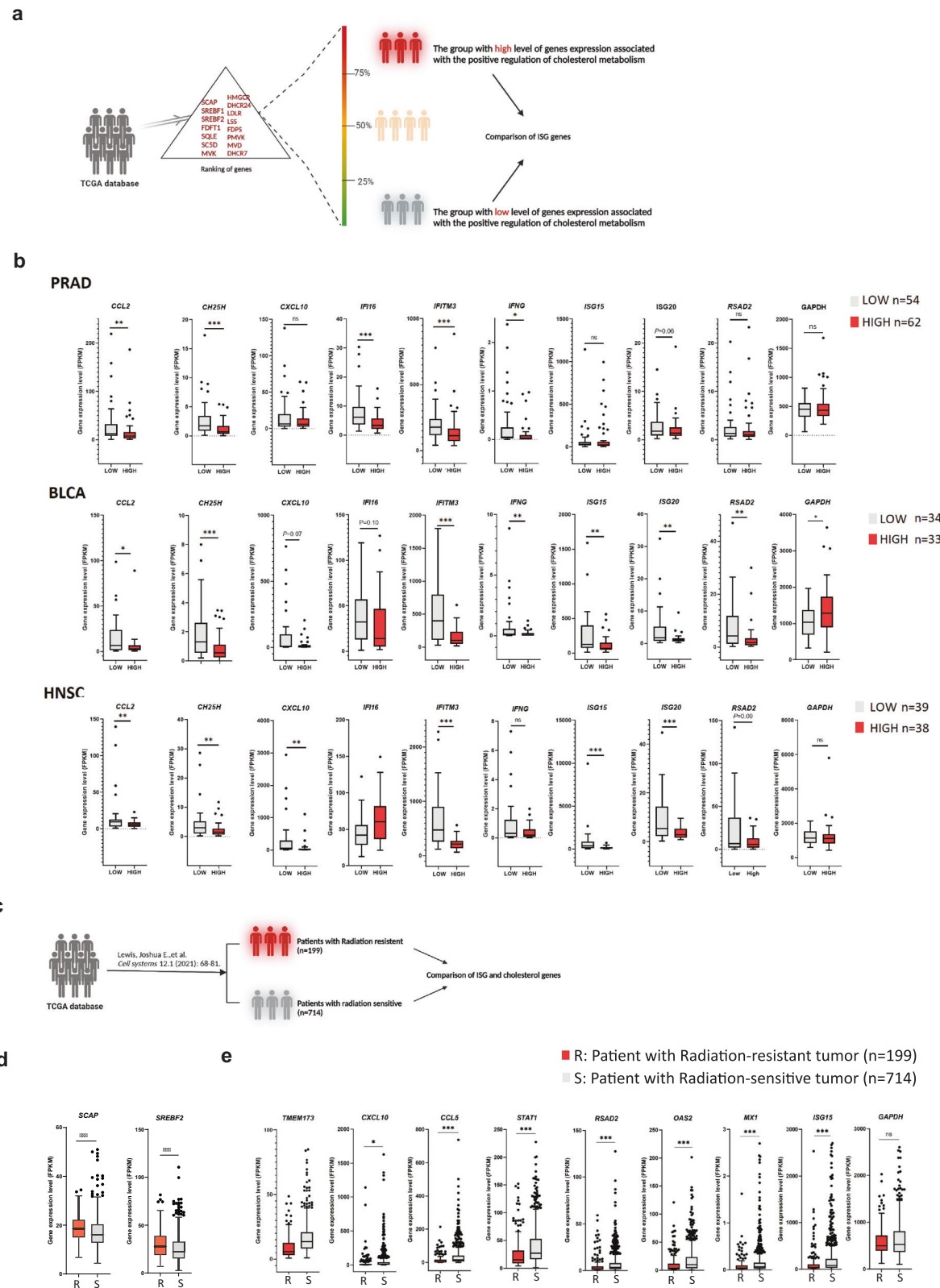

maximum sensitivity substrate (Thermo Fisher Scientific) or Clarity Western ECL substrate (BioRad). Antibodies used for Western blotting were rabbit mAb anti-phospho-STING (Ser366) (D7C3S) (1:1000, Cell Signaling Technology), rabbit mAb anti- STING (D2P2F) (1:1000, Cell Signaling Technology), rabbit mAb anti-phospho-TBK1/NAK (Ser172) (D52C2) (1:1000, Cell Signaling Technology), rabbit polyclonal anti-TBK1/NAK (D1B4) (1:1000, Cell Signaling Technology), rabbit anti-ABCG1 (ab52617) (1:1000, abcam), sheep anti-STING (AF6516) (1:1000, R&D Systems), rabbit anti-LSS (13715-1-AP) (1:1000, Proteintech), rabbit anti-STEEP (24021-1-AP) (1:1000, Proteintech), rabbit anti-SEC24 (D9M4N) (1:1000, Cell Signaling Technology), rabbit anti-IRF3 (D83B9) (1:1000, Cell Signaling Technology), rabbit anti-hospho-IRF3 (Ser386) (E7J8G) (1:1000, Cell Signaling Technology), rabbit anti-ERGIC53/LMAN1 (E2B6H) (1:1000, Cell Signaling Technology), rabbit

**Fig. 6 | The expression level of genes involved in positive cholesterol metabolism is inversely correlated with the response of ISGs in cancer.**
**a** Classification of TCGA patient tumors based on expression level of the indicated genes related with positive cholesterol regulation. **b** Box plots of The Cancer Genome Atlas (TCGA) RNA expression profiles in prostate adenocarcinoma (PRAD), Bladder Urothelial Carcinoma (BLCA), and Head-Neck Squamous Cell Carcinoma (HNSC). The highest and lowest 25% of cholesterol metabolism were analyzed by comparing cholesterol metabolism-high and cholesterol metabolism-low groups, respectively. Statistical analysis was performed using a two-tailed Mann-Whitney test. The upper and lower ends of the boxes represent the upper and lower quartiles, and the horizontal line inside the box is the median of the dataset. The whiskers indicate the upper and lower extremes of the dataset ($^{NS}p > 0.05$, $**p < 0.01$). **c** Classification of TCGA patient tumors into radiation-sensitive and -resistant classes based on the previous report[51]. **d, e** Box plots of TCGA RNA expression profiles are shown in panel (**d**) for cholesterol synthesis genes SCAP and SREBF1, and in panel (**e**) for ISG genes, comparing the radiation-sensitive and radiation-resistant classes. (ns not significant; $*p < 0.05$; $**p < 0.01$; $***p < 0.001$).

anti-STING (D1V5L) (1:1000, Cell Signaling Technology), rabbit anti-phospho-STING (Ser365) (D8F4W) (1:1000, Cell Signaling Technology), mouse mAb anti-Vinculin (hVIN1, 1:10000) (Sigma–Aldrich). The primary antibody dilution for all experiments, unless otherwise specified, was 1:1000. Secondary antibodies were donkey IgG anti-mouse-HRP (1: 10000, Jackson Immunoresearch) and donkey IgG anti-rabbit-HRP (1:10000, Jackson Immunoresearch).

### Confocal microscopy
For the STING trafficking, THP1 cells were treated with 2 mM MβCD for 2 h and then stimulated with or without 50 μg/ml cGAMP in medium for 60 min. The cells were fixed through pre-cold methanol for 5 min at −20 °C and then blocked in 1x PBS with 1% BSA for 1 h. Cells were stained with primary antibodies with sheep anti-STING (1:50 R&D Systems), mouse anti-PDI (1:100, Thermo), rabbit anti-GM130 (1:3000, Cell Signaling Technology) for 1 h and then stained using secondary antibody Donkey anti-Rabbit/Sheep/Mouse IgG (H + L) Highly Cross-Adsorbed Secondary Antibody, Alexa Fluor™ 488/568/647 (all 1:300, Alexa Fluor; Invitrogen) for 1 h. Images were taken with a Zeiss LSM 800 confocal microscope and processed with Zen software 3.8 (Zeiss). 10 images from each group were quantified through the Pearson correlation coefficient (r) for the STING-ER colocalization and the Manders' overlap coefficient for the STING-Golgi colocalization in ImageJ.

For the cholesterol and ER colocalization. THP1 cells were pre-stained with 1 μM ER-Tracker™ Red (BODIPY™ TR Glibenclamide) (Thermo) and 5 μM BODIPY 480/508-Cholesterol (Cayman Chemical) overnight and then treated with 50 μg/ml cGAMP for 0, 1, 2, and 4 h. The cells were fixed through with 4% formaldehyde for 10–20 min at 37 °C. Images were taken with a Zeiss LSM 800 confocal microscope and processed with Zen software 3.8 (Zeiss). 15 images from each group were quantified through the Pearson correlation coefficient (r) in ImageJ.

To measure the Sec24 spots per cell, THP1 cells were treated with vehicle, 5 mM Cholesterol overnight, or 2 mM MβCD for 2 h. The cells were fixed through pre-cold methanol for 5 min at −20 °C and then blocked in 1× PBS with 1% BSA for 1 h. Cells were stained with primary antibodies with rabbit anti-Sec24 (1:100, D9M4N, CST), and mouse anti-PDI (1:100, Thermo, Cat# MA3-019) overnight at 4 °C and then stained using secondary antibody (all 1:300, Alexa Fluor; Invitrogen) for 1 h. Images were taken with Zeiss LSM 800 confocal microscope and processed with Zen software 3.8 (Zeiss). The spots measurements of Sec24 were performed using the morphoLibJ plugin (segmentation-morphological segmentation) and analyze particles in ImageJ.

To analyze the correlation of ER membrane curvature and STING, HaCat cells were transfected with GFP[133] for 24 h. After this, the cells were treated with vehicle or 2 mM MβCD for 2 h and then stimulated with vehicle or 50 μg/ml cGAMP in medium for 40 min. The cells were fixed through pre-cold methanol for 5 min at −20 °C and then blocked in 1× PBS with 1% BSA for 1 h. Cells were stained with primary antibodies with sheep anti-STING, and mouse anti-PDI overnight at 4 °C and then stained using secondary antibody (all 1:300, Alexa Fluor; Invitrogen) for 1 h. Images were taken with a Zeiss LSM 800 confocal microscope and processed with Zen software 3.8 (Zeiss).

To measure the area of Tubular ER and Flat ER, THP1 cells were treated with vehicle or 2 mM MβCD for 2 h and then stimulated with vehicle or 50 μg/ml cGAMP in medium for 40 min. The cells were fixed through pre-cold methanol for 5 min at −20 °C and then blocked in 1× PBS with 1% BSA for 1 h. Cells were stained with primary antibodies with sheep anti-STING (R&D Systems, AF6516), mouse anti-Climp63 (1:100, Enzo Life Sciences, Cat#ENZ-ABS669), rabbit anti-Nogo (1:100, Abcam, ab47085) overnight at 4 °C and then stained using secondary antibody (all 1:300, Alexa Fluor; Invitrogen) for 1 h. Images were taken with a Zeiss LSM 800 confocal microscope and processed with Zen software 3.8 (Zeiss). Area measurements of Flat ER and Tubular ER were performed using the morphoLibJ plugin in ImageJ. Following the collection of area data, a ratio of Tubular ER to Flat ER was calculated within the same cell.

### ImageStream
Image Stream analysis was applied to investigate STING-ER colocalization. HEK293T cells with stable Flag-STING expression were pre-treated with vehicle or 2 mM MβCD for 1 h and stimulated with vehicle or cGAMP (50 ug/ml) for 1 h. The cells were fixed with 4% PFA for 15 min at room temperature, and then pre-permeabilized with 0.2% Triton X-100 for 6 min. The cells were incubated with mouse anti-PDI (ER marker), sheep anti-STING, and rabbit anti-GM130 (Golgi marker) for 1 h on ice, and then incubated with the Alexa-Fluor-labeled secondary antibodies for 1 h. Data was acquired through the Amnis ImageStream imaging flow cytometry (Luminex) and the level of colocalization was determined using IDEAS 6.2. Three sample replicates were included under the same conditions, with 10,000 cells per sample collected to calculate the colocalization percentage.

For ER membrane curvature assay, GFP-tagged Amphipathic lipid packing sensor motifs (ALPS) (GFP[133], the marker of ER membrane curvature) was transfected into HEK293T-STING (with a stable integration of the STING gene for expression) for 24 h. Then, the cells were pre-treated with vehicle or 2 mM MβCD for 1 h and stimulated with vehicle or cGAMP (50 ug/ml) for 1 h. The cells were fixed with 4% PFA for 15 min at room temperature and then pre-permeabilized with 0.2% Triton X-100 for 6 min. The cells were incubated with primary antibodies mouse anti-PDI for 1 h on ice and then incubated with the Alexa-Fluor-labeled secondary antibodies for 1 h. Data was acquired through the Amnis ImageStream imaging flow cytometry (Luminex) and the level of colocalization was determined using IDEAS 6.2. Three sample replicates were included under the same conditions, with 10,000 cells per sample collected to calculate the colocalization percentage.

For Cholesterol-ER colocation, THP1 cells were pre-stained with 0.5 μM TopFluor^R Cholesterol and 1 μM ER-Tracker™ Red overnight, and then treated with 2 mM MβCD or stimulated with cGAMP. Cells were fixed using 4% PFA for 15 min at RT. Cells were analyzed using Amnis ImageStream imaging flow cytometry (Luminex). Data were analyzed by IDEAS 6.2. Three sample replicates were included under the same conditions, with 10,000 cells per sample collected to calculate the colocalization percentage.

### In vitro budding assay
The reaction was conducted in accordance with our previously established protocol[42]. To isolate the cytosol, THP1 cells were first homogenized in a B88 buffer (comprising 20 mM HEPES-KOH, pH 7.2, 250 mM sorbitol, 150 mM potassium acetate, and 5 mM magnesium

acetate), supplemented with protease inhibitors. The cell homogenization process involved passing the mixture through a 25 G needle 20 times, followed by subjecting the homogenates to four rounds of centrifugation at 160,000 $g$ for 30 min. The resulting clarified supernatant fractions were collected and designated as the cytosol fraction. For the membrane fraction, THP1 cells were treated with or without MβCD (2 mM) for 4 h. Subsequently, the cells were lysed in a buffer containing 20 mM HEPES-KOH, pH 7.2, 400 mM sucrose, and 1 mM EDTA. The homogenates were initially centrifuged at 1000 $g$ for 10 min, after which the supernatant was further subjected to centrifugation at 100,000 g for 1 h. Following centrifugation, the resulting pellets were reconstituted in B88 buffer adjusted to an optical density at 600 nm ($OD_{600\ nm}$) of 10, constituting the total membrane fraction. The budding reaction was executed at 30 °C for 1 h, utilizing membranes with an $OD_{600\ nm}$ of 10, THP1 cytosol (final concentration of 2 mg/ml), and GTP (0.15 mM), supported by a functional ATP regeneration system (consisting of 40 mM creatine phosphate, 0.2 mg ml−1 creatine phosphokinase, and 1 mM ATP). After the reaction, the solution was centrifuged at 20,000 $g$ for 20 min, and the collected supernatant was then subjected to further centrifugation at 100,000 g for 30 min. The resulting pellets were dissolved in a 2× SDS reducing sample preparation buffer for subsequent immunoblotting analysis. It's important to note that all the above centrifugation steps were carried out at 4 °C.

### Cholesterol beads pull-down
To extract Flag-STING WT/R78A, Y81A and V85A/K150A, F153A, and V155M proteins, HEK293T cells were lysed in IP lysis buffer containing Protease Inhibitor Cocktail and PhoSTOP (Sigma). After centrifugation at 14,000 $g$ for 10 min at 4 °C, the cleared cell lysates were mixed with FLAG® M2 Magnetic Beads (Sigma) and incubated for 2 h at room temperature. The immunoprecipitated complexes were eluted with 3X FLAG® tag peptide (Sigma–Aldrich) in the wash buffer. Then, 3 µg elutes from cells with Flag-STING WT/Mutants were added to 50 µl cholesterol-coated beads (Echelon Biosciences Inc.) and another 3 µg elutes from cells with Flag-STING WT were added to 50 µl blank-coated beads as control and incubated for 1 h at room temperature. The beads were then washed five times with a buffer containing 10 mM HEPES, pH 7.4, 150 mM NaCl, and 0.5% Igepal. Following the last wash, the eluted bound proteins were analyzed through Western Blot by adding an equal volume of 2× Laemmli sample buffer to beads and heating them to 95 °C for 5–10 min.

### Mass spectrometry
To identify and quantify cholesterol binding on Flag-tagged STING WT or mutants, a liquid chromatography-tandem mass spectrometry method was developed. An ExionLC™ Series UHPLC was connected to a Sciex QTRAP 6500+ mass spectrometer with electrospray ionization (ESI) in positive mode. Compound detection was optimized by introducing the cholesterol reference standard to an electrospray source through a syringe pump, and the optimal mode for maximum sensitivity was determined using Q1MS scan mode. The following compound-dependent MS parameters were optimized: de-clustering potential (DP), entrance potential (EP), collision energy (CE), and collision cell exit potential (CXP). Instrument control and data acquisition were performed using Analyst Software (version 1.6.2), and quantifications were done by MultiQuant (version 3.0.3) based on a standard curve. Separation was performed using a reversed-phase ACQUITY UPLC CSH C18 (2.1 mm × 100 mm, 1.7 µm) with the mobile phase composed of a binary solvent mixture of solvent A (60% Acetonitrile, 40% H2O, 0.1% formic acid, and 5 mM ammonium formate) and solvent B (90% isopropanol, 10% acetonitrile, 0.1% formic acid, and 5 mM ammonium formate). The flow rate was set at 0.2 mL/min with an injection volume of 10 µL. The LC conditions were optimized with a column oven temperature set at 50 °C, curtain gas set at 20 psi, ion spray voltage set at

−4500 V, temperature set at 500 °C, ion source gas 1 set at 60 psi, and ion source gas 2 set at 50 psi. The binary gradient was set as follows: 0–2 min for column equilibration with solvent A at 60%, 2–2.1 min for ramping to 50% A, 2.1–12 min for ramping to 46% A, 12–12.1 min for ramping to 30% A, 12.1–18 min for ramping to 1% A, 18–21 min for isocratic hold with 1% A, and 21.1–23 min for hold with 60% A. Data points of the standard curves were weighted according to x-1.

### Statistics & reproducibility
The sample size, number of trials, and statistical methods used for each experiment have been provided in the respective figure captions. Data were analyzed with GraphPad Prism 9.0 (GraphPad Software). In cell experiments, comparison between two groups of data was performed using a two-tailed $T$-test, while comparison of multiple groups was conducted using one-way ANOVA with Tukey's multiple comparison test. In mouse experiments, statistical analysis of tumor growth utilized two-way ANOVA with Geisser-Greenhouse correction followed by Holm–Sidak's multiple comparisons test. Survival analysis in mice was performed using the Mantel-Cox Log-rank test.

### Reporting summary
Further information on research design is available in the Nature Portfolio Reporting Summary linked to this article.

## Data availability
The datasets generated during and/or analyzed during the current study are available from the corresponding authors on a reasonable request. Source data are provided with this paper.

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

## Acknowledgements

We thank Ane Kjeldsen for technical assistance in the laboratories and the animal facility at Department of Biomedicine for supporting mouse handling and Brita Holst Serup for technical assistance in the laboratories at Department of Health Science and Technology, Aalborg University. Image stream was performed at the FACS Core Facility, Aarhus University, Denmark, and confocal imaging was performed at the Bioimaging Core Facility, Aarhus University, Denmark. Images embedded in figures were generated using Biorender.com software. Funding was received from European Research Council (S.P.R.; ERC-AdG ENVISION; 786602); The Novo Nordisk Foundation (S.P.R.: NNF18OC0030274 NNF20OC0063436); The Danish Cancer Society (E.K.O.: R149-A10193-16S-47), Aase and Einar Danielsens Foundation (E.K.O.); The Erichsen Family Foundation (E.K.O.); The Lundbeck Foundation (M.R.J.; R238-2016-2708) and The Novo Nordisk Foundation (M.R.J.; NNF20OC0062825).

## Author contributions

B.C.Z., S.R.P., E.K.O. and M.R.J. conceptualized the project. B.C.Z., S.P.R., M.F.L., A.E., E.N., E.K.O., and M.R.J. designed the experimental setups, and B.C.Z., M.F.L., L.H., H.H., R.N., A.S.H., Y.Z., M.M., A.B., and A.E. performed the experiments with support from E.N., A.G., C.Z., and M.J. J.H. and X.D. performed the bioinformatic analysis. L.S.J., A.S.H., A.E., E.K.O., and M.R.J. designed and conducted animal experiments. B.C.Z., T.H.M., E.K.O., S.R.P., and M.R.J. supervised the experiments. R.A. provided invaluable resources and support of the DC-related experiments. B.C.Z., M.F.L. drafted the manuscript and E.K.O., S.R.P., and M.R.J. reviewed and edited the manuscript. All authors reviewed, edited, and approved the final manuscript.

## Competing interests

M.R.J. is shareholder within STipe Therapeutics that develop cancer immunotherapies targeting the STING pathway. The remaining authors declare no competing interests.
