## [Peer Review File · Nature Communications]

Cholesterol-binding motifs in STING that control endoplasmic retention mediate anti-tumoral activity of cholesterol-lowering compoundsEditorial Note: Parts of this Peer Review File have been redacted as indicated to remove third-party material where no permission to publish could be obtained.

REVIEWER COMMENTS

Reviewer #1 (Remarks to the Author):

In this study, the authors show that ER cholesterol sets a threshold for STING activation through two potential cholesterol recognition/interaction amino acid consensus (CRAC) motifs of STING. Upon cGAMP stimulation, the cholesterol levels within ER decline transiently presumably by SOAT1-dependent esterification, which induces ER membrane curvature to promote STING departure from ER, thus potentiating its activation. In mouse models, intratumoral cholesterol depletion by methyl- β -cyclodextrin (M β CD) enhances STING-dependent anti-tumor responses, which, in combination with anti-PD-1 antibodies, suppresses tumor progression. These observations are interesting and may have clinical potential. However, normal distribution of cholesterol among cellular organelles is essential for mammalian cells. The OSBP drives cholesterol/PI4P exchange between the endoplasmic reticulum (ER), the main site for lipid synthesis, and the trans-Golgi network (TGN) and cholesterol export from ER to plasma membrane is vital for the establishment of a cholesterol gradient along organelles of the secretory pathway. Cholesterol depletion leads to impaired cholesterol/PI4P exchange and collapse of the secretory pathway. cGAMP-activated STING exits from ER and translocates to the Golgi apparatus, endosomes, and finally to lysosomes. Previous work showed that either increasing (PMID: 26686653) or depleting cellular cholesterol level (34099821; 36921576) led to inhibited STING activation, probably due to disturbed STING trafficking. Disturbance of sphingomyelin, another major phospholipid in mammalian membranes had similar effects on STING activation. It's hard to perceive that cellular cholesterol binding compounds methyl- β -cyclodextrin or filipin III promoted STING activation by depleting cellular cholesterol. In line with this, the overall cholesterol-depletion-promoted STING activation under cGAMP stimulation in most, if not all, of experiments is very subtle or even negligible. In addition, this work is very descriptive and preliminary, more mechanistic studies are required. The manuscript was not well written, especially in the figures, missing a lot of labels and information, very hard to understand presented results.

Minor points:

1. Fig.1 showed that neither cholesterol depletion nor LSS KO did induce STING auto-activation. However, Fig 3a and Fig 4e-g showed that cholesterol depletion or mutation of CRAC motif in STING led to STING auto-translocation. What caused the inconsistencies? In Figure 1(c), more data is required to verify LSS /ABCG1 KO have an impact on STING activation in a cholesterol dependent manner.

1) Detecting the changes of cholesterol level on ER in LSS /ABCG1 KO cells.

2) Design two other control groups: LSS KO cells plus cholesterol and ABCG1 KO cells plus M β CD.

2. Fig 2a, 2b need more cells or experiments to support the authors' conclusion. The authors showed that cGAMP stimulation decreased cholesterol-ER colocalization. But what's confusing is how cGAMP affects ER cholesterol levels in a SOAT1-dependent manner, as cGAMP is a second messenger specifically activates STING. Have you ever tested changes of cholesterol-ER colocalization after cGAMP stimulation in STING KO cells to find out if STING is involved in this process? More experiments are needed to elucidate the relationship between cGAMP, STING, SOAT1 and the dynamic changes of ER cholesterol level. Further, in Fig 2c, it appeared there is nothing in OSBP KO cell sample.

3. The authors claimed that decreased cholesterol level upon cGAMP stimulation is mediated by SOAT1. But in Fig. 2c, 2d:

1) Almost in all KO cells, STING activation was partially affected. How to explain the function of other genes?

2) The influence of SOAT1 KO on ER cholesterol level in Fig. 2d didn't seem to match the degree of influence it had on STING activation in Fig. 2c.

3) Lack of the cGAMP untreated control in Fig. 2c.

4) The sample loading of OSBP KO group is obvious lower than others.

4. As mentioned in point #1, Fig. 3a showed that M β CD treatment alone significantly caused STING Golgi localization ($P < 0.01$), but none of previous result showed that cholesterol depletion alone activated STING (Fig.1a, 1c, and S1a). How to explain this disagreement? Fig. 3c, it showed that M β CD treatment apparently altered global morphology of ER, especially with the combination of cGAMP. So how cholesterol depletion promotes STING ER exit given such disturbed ER appearance? Fig. 3d, it's very hard to tell difference among these groups.

5. Fig 4a, detailed images of ER are required to examine the curvature alteration caused by cholesterol depletion around STING loci. Fig 4e-g, are WT/Mut STING transiently or stably transfected? If they were transiently transfected, how much was transfected? How come the activation was so low?
6. The major conclusion "Cholesterol depletion facilitates STING trafficking from ER to Golgi" needs more evidence.

Reviewer #2 (Remarks to the Author):

Zhang et al describe in this work a possible molecular mechanism that links cholesterol with STING activation and its well know downstream effects in promoting ISG response and immune activation. They also provide evidence in vivo of the potential anti-tumour effects this pathway has.

The work is quite original, as it describes, for the first time, a molecular mechanism that might explain previous observations where by cholesterol depleted ER promotes Type I responses (York 2015). This makes the manuscript impactful. However, there are some major concerns that, in my opinion, need to be addressed before publication.

In general, the description and annotation of experiments, in both M&M section, results and figure legends is very poor, including some typos, and I provide some examples below. This makes the data interpretation of the work difficult, hindering considerably the robustness of the conclusions.

Experiments, especially western blots should be clearly labelled and explained. All legends should include more descriptive information: for western blots, were they repeated? If n=1 this should be clearly explained and justified. If n>1, is data shown a representative image? of how many independent measurements? Can repeats be quantified and plotted? When cumulative data is represented, what do bars and errors represent? Statistical approach should also be explained in all legends. Dosage and/or exposure time is missing in some cases.

Finally, some of the conclusions are over-stated (again, some examples below), specially as cholesterol depletion is not demonstrated, as no cholesterol measurements have been taken. Authors should either quantify cholesterol in the system or not assume that their MbCD treatment depletes cholesterol. Figure legends should be more precise (i.e. Fig1S) to reflect this.

Methods:

1. Some protocols are missing (human CD differentiation, CRISPR/Cas9 or the use of AAVS1 as control).
2. Ethical approval for the use of human samples should be clearly stated in the methods.
3. The use of cell material (cell lines or primary) is not clearly stated and justified in the results section. The origin of cell lines is not explained (where they purchased from a collection?) Neither is clear in the figure legends. ImageStream experiments use 293T cells, not mentioned in the cell culture section.

Results

Cholesterol impairs STING activation.

1. Authors perform the experiments in the THP-1 cell line. This should be stated and justified.
2. In Fig1S authors use human moDCs – could they explain in the results text? Is this done as a validation of their findings in a cell line (THP-1) into a primary human cell? And if so, why is the cholesterol depletion changed from MbCD to Filipin III? Could authors add a reference that demonstrates Filipin III interference with cholesterol? This reviewer is aware of Filipin III binding cholesterol properties, but could not find any work that describes how it can "interfere" with cholesterol in such a way that blocks its binding to STING (which is not shown here either).
3. Fig1b: Understanding the figure is difficult: do top and bottom images correspond to the same blot at different exposure? If this is the case, authors should indicate this. From this single image it

is unclear to me that differences are found in oligomer levels in the presence of MbCD a dose dependent increase. Could authors quantify and plot cumulative data of their experimental replicates? It also contains a typo ("STING oligomer").

4. Fig1c. The evidence of KO is not homogeneous: LSS protein levels seem really low, but this is not the case for ABCG1 (where there is clearly some level of protein detected). Did authors control their KO efficiency?

5. S1b and c: data that show same conditions among b and c do not show the same levels of IFN or CXCL10 (1mM MbCD + 5uM cGAMP shows detectable levels of the cytokine and chemokine in b but not in c). Can authors explain this discrepancy? Could they also indicate in their legend the number of replicates? What the cumulative data and error show?

6. S1d and e: evidence of STING KO?

7. S2b: could authors show a representative flow cytometry plots of the data? As they have done for other supplementary information? This will give a better idea of the fold change they represent. And also explain in the legend the number of donors the figure shows?

8. Could authors improve figure 1 legend:

- Legend title does not reflect the results shown. Data shows that under cholesterol-depletion conditions, STING activation is potentiated. I would suggest a change in legend title that reflects better the outcome of the data shown.

- cGAMP dose in 1a, 1b, 1c.

- Figure 1D: what dose of MbCD was used? Figure 1a show differences between 0.5 and 2mM.

Reduction of ER cholesterol content.

1. Fig2a: can authors annotate what the three examples shown of ISx image deck represent? 0, 2 and 4h top to bottom? What do bars and error represent? Would authors consider scaling y axis to 0?

2. Fig2b: could authors please define mock? As a control to reproduce the ISx experiments, it would have been optimal to show 4h data, where co-localisation should be similar to mock (0h?).

3. Fig2c: evidence of KO? Replicates? Protein content in OSBP is not comparable to the other THP1 KO lines, so its role cannot be discarded.

4. Fig2d. Why was 2h not chosen? Replicates? Stats? Would authors consider scaling y axis to 0?

STING colocalization in ER vs Golgi

1. Fig3a. where cells co-staining with all markers together? If so, a much more visual representation of the data would be to show, in the same cell, how co-localisation of STING with ER is lower when compared to Golgi. Would authors consider scaling y axis to 0 in all their graphs?

2. Fig 3b. Could authors provide a quantification? The visual difference is not very convincing. How many times was the experiment performed? What were the MbCD doses chosen here? This essay is not evidence of trafficking but accumulation of STING in the budding fraction.

3. Fig 3c. To this reviewer, the visual difference is not very convincing, especially when trying to assess STING depletion from ER (DPI) in MbCD treatment. Could authors maybe use cholesterol control as in Fig 3d?

Direct binding of cholesterol to STING.

1. Figure legend lacks information about these experiments, such as cGAMP dosage and exposure times.

2. The results of the Fig4a and b are overstated (lines 191 and 192), as the differences found in both ER positive curvature and tubular ER are dependent on MbCD exposure, but independent of cGAMP treatment.

In vivo tumour model:

Figure legends have much more detail. Fig4S legend needs correction: there are no panels e and f.

Cholesterol gene expression associations.

1. Here, authors relate gene expression in their 15 gene candidates as "cholesterol levels". This is not very accurate, as cholesterol levels were not measured. Moreover, although most of their genes belong to either the cholesterol biosynthetic pathway or positive transcriptional regulation (SREBP2), their upregulation might not reflect increased cholesterol levels, but pretty much the opposite (SREBP2 expression is triggered when intracellular - ER - cholesterol levels are low).

2. Fig7 data does not demonstrate blocking of innate immune responses, as adaptive immune cells

can also express chemokines and cytokines and can respond to type I IFN stimulation.

I also would request clarification on the following minor issues:

- Acronyms definition (ERGIC – line 60, ALPS – line 459)
- Line 70-72 lacks references – work describing dying tumour cells' DNA uptake by immune infiltrating cells and cGAS-STING activation.
- What do authors define as “proper immune cells”? (line 91) – this should be properly defined: infiltration, activation, etc. Also, it would clarify if STING activation is required in the tumour cell, the immune cell or both.
- Line 116: MbCD is not a cholesterol inhibitor (it does not inhibit cholesterol synthesis or import) but it is a cholesterol depletion agent (as authors correctly state in the results section – line 122). This should be corrected.

Espe Perucha
Senior Lecturer in Experimental Rheumatology
King's College London

Reviewer #3 (Remarks to the Author):

This is a very interesting and well-written study from Zhang et al establishing a direct effect of cholesterol on STING regulation via cholesterol binding motifs in the protein and changes in ER structure. Further, they showed the STING activating property of cholesterol depletion could enhance tumor control and survival in a mouse model, and that genes regulating cholesterol correlated with ISG induction in human cancer subjects. This study will be of significant interest to the field and has therapeutic implications. There are no major experimental deficiencies, but the description and presentation of the figures need work to improve clarity.

General:

- 1) In general, could you please spell out acronyms or full molecule names the first time they are used? For instance, SOAT1 in line 112. Spell out LSS before the sentence explaining it.
- 2) In multiple figures, it should be “colocalization”, not “colocation”.
- 3) In multiple figures, the y-axis should go down to 0. Otherwise, it gives a false impression of the magnitude of effect (2a, 2d, 3a, 4f, 4g)
- 4) In the image stream analyses, it is difficult to appreciate the point – of course cholesterol localizes with ER staining (2a). Are the 3 rows representing 3 different cell examples of the same condition or 3 different conditions? 3a and 4a have the same issues. This should be better described in the figure legends. In general, the confocal images provide similar information but are much clearer.
- 5) There are multiple instances of blots missing lanes or with inadequate loading (see below).
- 6) The methods diagrams are OK but need to be accompanied by better written descriptions in either figure legends or methods. For specifics, see below.
- 7) Any ideas on how cGAMP might activate SOAT1?
- 8) Any ideas why the low but not high dose M-beta-CD worked in the tumor models? This surprising finding should at least be mentioned in the discussion.

More specific:

- 9) In Figure S1a, there is an increase at 4 and 8 hours, but actually a decrease at 20 hours (line 126).
- 10) In Figure 1c the AAVS1 control needs more description and should be referenced.
- 11) The graphs in Figure S1b and S1c showing an effect on STING functional output should be included in the main report.
- 12) Almost the whole paper (Figs 1-4) is in THP1 and HEK293. It would be helpful to incorporate at least a sample of the primary cell work in the supplemental figures back into the main body of the paper.

- 13) Does exogenous cholesterol have any impact on the biochemical readouts in Figure 1?
- 14) In Figure 1d it looks like there is a band missing for the 2 ug MbCD. Is there another blot you could use?
- 15) The control for Fig S1d-e (IFN-a) should be indicated in the figure legend.
- 16) In Figure S2a the figure and legend suggest the MbCD was used alone rather than in combination as suggested on line 141.
- 17) What happened to the OSBP CRISPR lane in 2c? Is there another blot you could use? If these are bulk transfections, the amount of knockout should be shown at least in supplemental data.
- 18) In Figure 3b, what is M and UT? This should go in the figure legend. It doesn't look like there is any difference upon adding MbCD. The vesicle budding study is difficult to understand and does not add very much to the report. It needs to be better described or removed.
- 19) In Figure 3c, could you add a STING/PDI column?
- 20) The ALPS-GFP assay requires description. For the tubular ER study (fig. 4b) Did you use anti-RTN as indicated in the figure legend or Nogo, as written on the Figure? How exactly were these "counted" and the tubular to flat ratios derived?
- 21) The labeling of Figure 4c western blot is not clear. Is this the cholesterol bead ip? And blot with FLAG or with STING? Or a STING ip? Why is the first lane of input missing?
- 22) The generation of the STING mutants should be described. It's missing.
- 23) The individual spaghetti plots do not really add anything to 5b.
- 24) Where is the results description of 5f? It goes from 5e to 5g. The upper row of 5f is also not good quality.
- 25) In 5h, is the blue line superimposed on the red for most of it?
- 26) Do you have the Ns for Figure 6a?
- 27) Please keep the color coding consistent between the 6c diagram and 6d and e graphs. It's exactly flipped.
- 28) In the human subjects, is there any information on plasma cholesterol levels or statin use?

Minor:

- 1) Line 65: Rather than IFN-I, it would be better to write "type I IFN".
- 2) Line 94: bell "shaped" rather than bell curved.
- 3) Line 123: missing the word "using" between cholesterol and Methyl-beta-cyclodextrin.
- 4) Line 166 should be "exits" rather than "exiting".
- 5) Line 231: "promote" rather than "prone"?
- 6) Line 295: the CRAC motif in the connecting loop (in addition to transmembrane region) should also be mentioned.

Rebuttal letter.

We appreciate the time the reviewers have spent on our manuscript and for the overall positive input to our work. We have tried to accommodate all comments and suggestions by the reviewers. Importantly, we have made thorough modifications throughout to the manuscript to improve the language, figure legends and overall understanding. Please see below a point-by-point response to all three reviewers comments.

Reviewer #1 (Remarks to the Author):

In this study, the authors show that ER cholesterol sets a threshold for STING activation through two potential cholesterol recognition/interaction amino acid consensus (CRAC) motifs of STING. Upon cGAMP stimulation, the cholesterol levels within ER decline transiently presumably by SOAT1-dependent esterification, which induces ER membrane curvature to promote STING departure from ER, thus potentiating its activation. In mouse models, intratumoral cholesterol depletion by methyl- β -cyclodextrin (M β CD) enhances STING-dependent anti-tumor responses, which, in combination with anti-PD-1 antibodies, suppresses tumor progression. These observations are interesting and may have clinical potential. However, normal distribution of cholesterol among cellular organelles is essential for mammalian cells. The OSBP drives cholesterol/PI4P exchange between the endoplasmic reticulum (ER), the main site for lipid synthesis, and the trans-Golgi network (TGN) and cholesterol export from ER to plasma membrane is vital for the establishment of a cholesterol gradient along organelles of the secretory pathway. Cholesterol depletion leads to impaired cholesterol/PI4P exchange and collapse of the secretory pathway. cGAMP-activated STING exits from ER and translocates to the Golgi apparatus, endosomes, and finally to lysosomes. Previous work showed that either increasing (PMID: 26686653) or depleting cellular cholesterol level (34099821; 36921576) led to inhibited STING activation, probably due to disturbed STING trafficking. Disturbance of sphingomyelin, another major phospholipid in mammalian membranes had similar effects on STING activation. It's hard to perceive that cellular cholesterol binding compounds methyl- β -cyclodextrin or filipin III promoted STING activation by depleting cellular cholesterol. In line with this, the overall cholesterol-depletion-promoted STING activation under cGAMP stimulation in most, if not all, of experiments is very subtle or even negligible. In addition, this work is very descriptive and preliminary, more mechanistic studies are required. The manuscript was not well written, especially in the figures, missing a lot of labels and information, very hard to understand presented results.

We agree that this manuscript can be improved by further details embedded in the figures and figure legends. We hope that the reviewer will appreciate the extensive work we have done to improve the results, descriptions as well as adding more mechanistical understanding, which we overall believe aid the understandability of our manuscript.

Regarding the effect of cholesterol depletion, the reviewer argues that this is detrimental for the cell. We agree that there is a fine balance between complete removal of all cholesterol from cellular membranes, and then the more reduced lowering concentration of cholesterol we obtain with low concentrations of M β CD. This may also explain why others have found diverging mechanisms of cholesterol influencing STING activity.

Regarding the issue of cholesterol removal and its differential impact on STING, we have included a substantial sentence in the "Discussion" section. Please see line 329-345.

Minor points:

1. Fig.1 showed that neither cholesterol depletion nor LSS KO did induce STING auto-activation. However, Fig 3a and Fig 4e-g showed that cholesterol depletion or mutation of CRAC motif in STING led to STING auto-translocation. What caused the inconsistencies?

We appreciate the reviewer for raising this issue. There are two main reasons for the observed inconsistency. Firstly, it is attributed to technical differences; in Figure 1, we utilized the WB

detection method, while in Figure 4e-g, we employed ISRE promoter activity assays. These two techniques exhibit variations in sensitivity. Secondly, in Figure 1, THP1 cells expressed STING endogenously, and it's possible that more cholesterol depletion or LSS knockout may not be sufficient to fully drive STING activation, or it may occur at very low levels. In contrast, in Figure 4e-g, we used HEK293T cells with exogenously high STING expression, which constitutes a strong stimulus for the cells. Consequently, simple cholesterol depletion or mutation of the CRAC motif in STING is adequate to induce STING activation and translocation in this context.

In Figure 1(c), more data is required to verify LSS /ABCG1 KO have an impact on STING activation in a cholesterol dependent manner.

1) Detecting the changes of cholesterol level on ER in LSS /ABCG1 KO cells.

To accommodate this, we chose to generate new LSS/ABCG1 knockout THP1 cells using CRISPR-Cas9 (fig S1e, S1f and S3b). Here we can see that LSS KO significantly reduced the cholesterol level on ER, while the ABCG1 KO significantly increased the cholesterol level on ER. This became more pronounced when we added cGAMP stimulation into the system. This is now described in the result section line 144 to 152 and shown in new Fig S3b. Also the KO score for LSS and ABCG1 is shown in the new fig S1e.

2) Design two other control groups: LSS KO cells plus cholesterol and ABCG1 KO cells plus M β CD. In our study, we treated ABCG1 KO cells with M β CD. For the relevant data, please refer to the new figure S1f. We did not continue with the experiments of LSS KO cells plus cholesterol treatment because we could not show that cholesterol treatment significantly increases the influx of cGAMP into the cells, thereby introducing additional experimental variables that would compromise our ability to draw reliable conclusions (see rebuttal Figure R1 – not included in the manuscript).

Figure R1. Intracellular cGAMP levels were analyzed using an ELISA assay. THP1 cells were treated with either a vehicle, 2mM M β CD for 2 hours, or 5mM Cholesterol overnight, followed by stimulation with a vehicle or 50 μ g/ml cGAMP for 6 hours. The intracellular cGAMP was isolated and detected using the 2'3'-cGAMP ELISA Kit (Cayman Chemical, Cat.#501700) following the manufacturer's instructions.

Fig 2a, 2b need more cells or experiments to support the authors' conclusion.

We would like to emphasize that the data in figure 2a was done using image stream that included more than 10,000 cells per analysis. We believe this gives the statistical power needed. For figure 2b we have now analyzed more cell images and made the statistical analysis of the data – please see details in the new Fig. 2c.

The authors showed that cGAMP stimulation decreased cholesterol-ER colocalization. But what's confusing is how cGAMP affects ER cholesterol levels in a SOAT1-dependent manner, as cGAMP is a second messenger specifically activates STING.

1) Have you ever tested changes of cholesterol-ER colocalization after cGAMP stimulation in STING KO cells to find out if STING is involved in this process?

Please see the new fig. 2f, which test cGAMP stimulation in WT cells versus STING KO cells. Measuring the cholesterol-ER colocalization shows that wildtype cells stimulated with cGAMP have a clear drop in colocalization whereas this is not seen in STING KO cells, which behave similarly as WT not treated with cGAMP.

2) More experiments are needed to elucidate the relationship between cGAMP, STING, SOAT1 and the dynamic changes of ER cholesterol level.

We thank the reviewer for suggesting this. Please see our new figure 2f where we have expanded our confocal analysis to include ER-cholesterol in SOAT1 KO cells with and without cGAMP.

3) Further, in Fig 2c, it appeared there is nothing in OSBP KO cell sample.

We agree with the reviewer and have repeated all the western blotting with new KO cell line samples including the usage of two independent sgRNA per target gene, to address this point. Please review new fig. 2d.

The authors claimed that decreased cholesterol level upon cGAMP stimulation is mediated by SOAT1. But in Fig. 2c, 2d:

1) Almost in all KO cells, STING activation was partially affected. How to explain the function of other genes?

The authors agree that the previously shown data may be difficult to interpret. We have therefore repeated the experiment several times, using multiple gRNAs per gene. Although we initially found effects on STING signaling for several cholesterol-regulating genes [and some were consistent with previous reports e.g. Fang et al 2023], the one that is clearly the most reproducible and robust, and which can now be fully explained mechanistically in our system, is SOAT1. Therefore, please review our revised Fig 2d. These data show that the effect of cholesterol-modulation on STING signaling (pSTING, pTBK1, pIRF3) in our system is best explained by a role for SOAT1. As also touched upon in other parts of this letter and discussed in the revised manuscript, the role for cholesterol in STING signaling is complex and well-defined experimental conditions need to be used to study a given aspect of the biology.

Fang, Run, et al. "ARMH3-mediated recruitment of PI4KB directs Golgi-to-endosome trafficking and activation of the antiviral effector STING." Immunity 56.3 (2023): 500-515.

2) The influence of SOAT1 KO on ER cholesterol level in Fig. 2d didn't seem to match the degree of influence it had on STING activation in Fig. 2c.

These two figures derive from different experimental setup and the method for detection was different. Thus, we find it inappropriate to do a direct comparison between the results. Nevertheless, with our new results the data between 2d and 2e-f are supportive.

3) Lack of the cGAMP untreated control in Fig. 2c.

We thank the reviewer for highlighting this. We have now added the non-stimulated samples and redo the western blot, please see new Fig. 2d.

4) The sample loading of OSBP KO group is obvious lower than others.

Same to the above response, a new relative western blot is now included in the manuscript. Please see new Fig. 2d.

4. As mentioned in point #1, Fig. 3a showed that M β CD treatment alone significantly caused STING Golgi localization (P<0.01), but none of previous result showed that cholesterol depletion alone activated STING (Fig.1a, 1c, and S1a). How to explain this disagreement?

We appreciate the reviewer observations and will try to give a rational argument for the differences between the figures. Importantly, Fig. 1a, 1c, and S1a were performed using THP-1 cells and detected by Western blotting (WB). All the protein detections were based on endogenous levels. On the other hand, Fig. 3a was conducted using HEK293T cells stably expressing STING. The lack of consistency between these two results can be attributed to the following reasons: 1) The cellular systems and experimental methods used in these two experiments have completely different detection sensitivities; 2) HEK293T cells with stable overexpression of STING inherently put the cells in a stimulated state. Hence, even without cGAMP stimulation, it is possible for STING to be activated and translocated in the case of M β CD treatment alone.

Fig. 3c, it showed that M β CD treatment apparently altered global morphology of ER, especially with the combination of cGAMP. So how cholesterol depletion promotes STING ER exit given such disturbed ER appearance?

We ask the reviewer to please pay attention to Fig. 4. Here we try to clearly explain that M β CD removal of cholesterol leads to an increase in curvature of the endoplasmic reticulum (ER) membranes, resulting in an elevated proportion of tubular ER. This will certainly lead to morphological changes in the ER as seen in figure 3c.

Fig. 3d, it's very hard to tell difference among these groups.

We appreciate the input from the reviewer, and we have tried to improve the quality of the images. In our and other previous studies, it has been established/confirmed that the COPII complex mediates the trafficking of STING from the endoplasmic reticulum (ER) to the ER-Golgi intermediate compartment (ERGIC). To map this in our experimental setup, we used SEC24 as a marker for COPII transport vesicles and observed a significant increase in the number of SEC24-formed vesicles after cholesterol removal, while increasing cholesterol resulted in a significant decrease in the number of SEC24-formed vesicles. Please see our new fig. 3d which also include statistical analysis. This is further supported by the new fig. 3d that calculate Sec24 spots.

5. Fig 4a, detailed images of ER are required to examine the curvature alteration caused by cholesterol depletion around STING loci.

We appreciate the reviewer's suggestion, and we have addressed it accordingly. We have now incorporated high-resolution confocal images that specifically highlight STING and membrane curvatures. In these new confocal images, we observe a noticeable increase in both the intensity and the surface area of GFP¹³³ (an ER membrane curvature probe) following cholesterol depletion. Additionally, the STING foci are more evenly distributed along the curvature area. These data support our image stream results. Please see new fig. S4.

Fig 4e-g, are WT/Mut STING transiently or stably transfected?

- If they were transiently transfected, how much was transfected?
- How come the activation was so low?

We used lentiviral transduction to deliver STING WT/Mut into HEK293T cells, resulting in stable and overexpressed STING driven by a CMV promoter. The reason for the low activation level seen is that we did not employ any stimulation; instead, we compared the differences in ISRE activation and STING translocation between STING WT and mutant after stable overexpression.

6. The major conclusion "Cholesterol depletion facilitates STING trafficking from ER to Golgi" needs more evidence.

In the revised manuscript, we have employed rigorous and independent methods, including ImageStream, confocal microscopy, and in vitro assays, to thoroughly demonstrate the promotion of STING trafficking from ER to Golgi upon cholesterol depletion. We believe that with all the good suggestions from the three reviewers that has improved our manuscript and strengthened the results therein, gives us the evidence and justification for our major conclusion.

Reviewer #2 (Remarks to the Author):

Zhang et al describe in this work a possible molecular mechanism that links cholesterol with STING activation and its well know downstream effects in promoting ISG response and immune activation. They also provide evidence in vivo of the potential anti-tumour effects this pathway has.

The work is quite original, as it describes, for the first time, a molecular mechanism that might explain previous observations where by cholesterol depleted ER promotes Type I responses (York 2015). This makes the manuscript impactful. However, there are some major concerns that, in my opinion, need to be addressed before publication.

In general, the description and annotation of experiments, in both M&M section, results and figure legends is very poor, including some typos, and I provide some examples below. This makes the data interpretation of the work difficult, hindering considerably the robustness of the conclusions.

First of all, we thank the reviewer for the positive input to the overall study and our findings. We appreciate the comments regarding presentation and description of our data. We have tried to accommodate this by extensive review of all figure layouts, adding new and more clear data sets, rewording many sections and adding more details in the figure legends.

Experiments, especially western blots should be clearly labelled and explained.

All legends should include more descriptive information:

We appreciate the input from the reviewer and have corrected figure legends appropriately.

- for western blots, were they repeated? If n=1 this should be clearly explained and justified. If n>1, is data shown a representative image? of how many independent measurements?
Information has been included in all figure legends
- Can repeats be quantified and plotted?
We quantified all the blots related to the activation of the STING pathway through three independent repeated experiments. For blots where the differences were visually and markedly evident, including Fig1d and Fig4c, we did not perform quantification but included the second time repeats of the three independent experiments. For the blot related to STING oligomers (Fig 1b), due to difficulties in accurate quantification, we decided to display the results from the three independent repeat experiments in our supplementary figures.
- When cumulative data is represented, what do bars and errors represent? We have added information in all figure legends when relevant
- Statistical approach should also be explained in all legends. We have added information in all figure legends when relevant
- Dosage and/or exposure time is missing in some cases. This has been corrected

Finally, some of the conclusions are over-stated (again, some examples below), specially as cholesterol depletion is not demonstrated, as no cholesterol measurements have been taken.

Authors should either quantify cholesterol in the system or not assume that their M β CD treatment depletes cholesterol. Figure legends should be more precise (i.e. Fig1S) to reflect this.

We understand the reviewer argument. To show that we do achieve cholesterol depletion we have now stained for cholesterol in cells treated with M β CD. This is for example demonstrated in new Fig. S1a-b.

Methods:

1. Some protocols are missing (human CD differentiation, CRISPR/Cas9 or the use of AAVS1 as control). We thank the reviewer for highlighting this. The human dendritic cell differentiation is mentioned in the first method section but has now been expended with more details. The details on CRISPR editing have been added.

2. Ethical approval for the use of human samples should be clearly stated in the methods. An ethical section has been added, however the blood samples we achieve from the local blood bank is not subjected to a specific project ethical approval in Denmark.

3. The use of cell material (cell lines or primary) is not clearly stated and justified in the results section. The origin of cell lines is not explained (where they purchased from a collection?) Neither is clear in the figure legends. ImageStream experiments use 293T cells, not mentioned in the cell culture section. We have added information throughout the manuscript to address this critic

Results

Cholesterol impairs STING activation.

1. Authors perform the experiments in the THP-1 cell line. This should be stated and justified done

2. In Fig1S authors use human moDCs – could they explain in the results text? We have added a sentence to highlight the use of primary human cell products

- Is this done as a validation of their findings in a cell line (THP-1) into a primary human cell? And if so, why is the cholesterol depletion changed from M β CD to Filipin III?

We have added a few sentence of this in the manuscript, but yes the idea was to confirm the phenomenon in other setting, regarding both cell type and drug that we know interfere with cholesterol. In supplement figure 2 (now Fig 1g and Fig S1c) we did a comparison of filipin-III and M β CD in primary human moDC to show that they feed into same mechanisms

- Could authors add a reference that demonstrates Filipin III interference with cholesterol? This reviewer is aware of Filipin III binding cholesterol properties, but could not find any work that describes how it can “interfere” with cholesterol in such a way that blocks its binding to STING (which is not shown here either).

We agree with the reviewer that Filipin III is not described in the literature as a cholesterol depleting drug but specifically binds strongly to cholesterol molecules in membranes. We used this in our work as we from an initial screening of different compounds showed that Filipin-III had similar effects as M β CD when treating human moDCs with drugs and cGAMP (see Figure R2 – not included in the manuscript). We are willing to omit the data if necessary as we do not understand the true mechanism of how Filipin-III works in interfering STING and cholesterol.

Figure R2. Flow cytometry analysis of HLA-DR and CD86 expression in human moDCs pre-treated with amiloride (ami), chlorpromazine (cpz) or filipin-III (fili) prior to 24 hrs stimulation with cGAMP. Statistical significance was calculated using one-way ANOVA with Turkey’s multiple comparison correction (****, $p < 0.0001$)

3. Fig1b: Understanding the figure is difficult:

do top and bottom images correspond to the same blot at different exposure? If this is the case, authors should indicate this.

For this experiment, the top and bottom images were not from the same blot. For transparency we have added the two other independent repeated experiments. Please see Fig S1e.

From this single image it is unclear to me that differences are found in oligomer levels in the presence of M β CD a dose dependent increase. We appreciate the input from the reviewer. We have tried to make the figure clearer as to what the different bands represent. This is also added in the result section see line 138-141.

Could authors quantify and plot cumulative data of their experimental replicates? It also contains a typo ("STING oligomer"). For the blot related to STING oligomers (Fig 1b), due to difficulties in accurate quantification, we decided to display the results from the three independent repeat experiments in our supplementary data – see Fig S1e.

4. Fig1c. The evidence of KO is not homogeneous: LSS protein levels seem really low, but this is not the case for ABCG1 (where there is clearly some level of protein detected). Did authors control their KO efficiency? We have now included the KO efficiency. Please see the Fig S2a.

5. S1b and c: data that show same conditions among b and c do not show the same levels of IFN or CXCL10 (1mM M β CD + 5uM cGAMP shows detectable levels of the cytokine and chemokine in b but not in c).

Please be aware that this figure is now represented as Fig 1e and f

- Can authors explain this discrepancy? We agree with the reviewer that the data from these two setups are not directly comparable. These reflect that experiments were done at different time points and show some assay-to-assay differences. However, the conclusion remains in support of the combination of M β CD and cGAMP gives a stronger cytokine induction.
- Could they also indicate in their legend the number of replicates? We have indicated the number of replicates in legend (new Fig 1e, 1f).
- What the cumulative data and error show? Data are shown as mean \pm SD value of three replicates.

6. S1d and e: evidence of STING KO? The cells used have been described in a prior publication (Prabakaran et al., 2018). Please see below the western blot from this paper.

[REDACTED]

Prabakaran, T., Bodda, C., Krapp, C., Zhang, B. C., Christensen, M. H., Sun, C., ... & Paludan, S. R. (2018). Attenuation of cGAS-STING signaling is mediated by a p62/SQSTM1-dependent autophagy pathway activated by TBK1. *The EMBO journal*, 37(8), e97858.

7. S2b: could authors show a representative flow cytometry plots of the data? As they have done for other supplementary information? This will give a better idea of the fold change they represent. And also explain in the legend the number of donors the figure shows? These data are now moved to primary fig 1g. The flow plot, gating and histograms are now included as new fig. S2

8. Could authors improve figure 1 legend:

- Legend title does not reflect the results shown. Data shows that under cholesterol-depletion conditions, STING activation is potentiated. I would suggest a change in legend title that reflects better the outcome of the data shown. We agree and have made proper correction.

- cGAMP dose in 1a, 1b, 1c. The dose information has been added in the figure legend.
- Figure 1D: what dose of M β CD was used? Figure 1a show differences between 0.5 and 2mM. This has been added in the figure legend

Reduction of ER cholesterol content.

1. Fig2a: can authors annotate what the three examples shown of ISx image deck represent? 0, 2 and 4h top to bottom? What do bars and error represent? Would authors consider scaling y axis to 0?

We appreciate the input from the reviewer on how to improve this figure panel. In order to enhance the understanding of the data, we have now included labels and specific details in both the figure and figure legend. In the experiment, we conducted three sample replicates, collecting 10,000 cells per sample to calculate the co-localization percentage. In the Figure, the bars represent the average percentage of co-localization from three samples, and the error bars indicate the standard deviation of three samples. We have re-scaled the y axis to 0.

2. Fig2b: could authors please define mock? As a control to reproduce the ISx experiments, it would have been optimal to show 4h data, where co-localisation should be similar to mock (0h?).

We appreciate the comment and we have now updated the figure to also include 4 hrs. Please see new fig. 2b-c. The mock sample is described as "untreated (mock)"

3. Fig2c: evidence of KO? Replicates? Protein content in OSBP is not comparable to the other THP1 KO lines, so its role cannot be discarded. We have repeated the KO samples and generated a new western blot where samples are more comparable. Please see the new Fig 2d.

4. Fig2d. Why was 2h not chosen? Replicates? Stats? Would authors consider scaling y axis to 0?

We understand why the reviewer asked and we went back and reviewed our protocols for correctness. After verification, we found that the "1h" in the Figure legend is incorrect, and the accurate time point is "2h". This has now been changed accordingly. Furthermore, in our study, we conducted three sample replicates, collecting 10,000 cells per sample to calculate the co-localization percentage. To provide a more straightforward representation of the differences observed after SOAT1 knockout, we changed the cell system from SOAT1 KO THP1 cells created through electroporation delivery into SOAT1 KO THP1 cells created by the lentivirus delivery. The latter one was selected through puromycin that can make stable SOAT1 knockout and keep higher proportion of SOAT1 ko cells. Through this approach, we remove the background as much as possible and received data with scaling y axis started from 0. To bolster the robustness of our conclusions, we have also included additional confocal data, which presents a more pronounced and visually intuitive differentiation. Please refer to the new Fig. 2f.

STING colocalization in ER vs Golgi

1. Fig3a. where cells co-staining with all markers together? If so, a much more visual representation of the data would be to show, in the same cell, how co-localisation of STING with ER is lower when compared to Golgi.

We have updated our representation by co-localizing all markers within the same cell.

In fact, we expected to observe that the reduction in STING-ER co-localization would correspondingly increase STING-Golgi co-localization with same level. However, at our current technical level of ImageStream, it is still a challenge. The reasons for these differences may include variations in the binding affinity of ER and Golgi marker antibodies. Additionally, staining disparities may arise from differences in the secondary antibodies for fluorescence detection, as well as variations in intensity compensation between different fluorescence channels.

Would authors consider scaling y axis to 0 in all their graphs?

Due to the limited resolution of the imaging system and the widespread distribution of the endoplasmic reticulum (ER) within the cells, we encountered relative high background for the ER-STING colocalization. However, we were still able to observe significant differences between the various groups. Therefore, for visual interpretation we would argue against scaling the y-axis to 0, but we have now modified the graphs.

2. Fig 3b. Could authors provide a quantification? The visual difference is not very convincing. How many times was the experiment performed? What were the M β CD doses chosen here? This essay is not evidence of trafficking but accumulation of STING in the budding fraction.

We have optimized and re-done this experiment, including a new western blot with different M β CD dosages, which is now incorporated into new Fig 3b.

We agree that "This essay is not evidence of trafficking." In order to mitigate potential complexities within the cellular system, we aimed to validate, through a straightforward in vitro assay, whether M β CD can promote the formation of STING budding vesicles. Our previous work and studies from other laboratories have already confirmed the necessity of COPII budding vesicles for STING's migration from the ER. Furthermore, this budding assay has been employed in previous studies to demonstrate "Budding of COPII Vesicles of ER-to-Golgi Trafficking." (Joo et al., 2016; Ge et al., 2013). Therefore, while this experiment may not directly prove STING migration from the ER to the Golgi, it serves as supporting evidence to indicate that cholesterol depletion facilitates the formation of STING-COPII budding vesicles.

Joo, J. H., Wang, B., Frankel, E., Ge, L., Xu, L., Iyengar, R., ... & Kundu, M. (2016). The noncanonical role of ULK/ATG1 in ER-to-Golgi trafficking is essential for cellular homeostasis. *Molecular cell*, 62(4), 491-506.

Ge, L., Melville, D., Zhang, M., & Schekman, R. (2013). The ER-Golgi intermediate compartment is a key membrane source for the LC3 lipidation step of autophagosome biogenesis. *elife*, 2, e00947.

3. Fig 3c. To this reviewer, the visual difference is not very convincing, especially when trying to assess STING depletion from ER (DPI) in M β CD treatment. Could authors maybe use cholesterol control as in Fig 3d? To address the reviewer concern, we have now added quantification of n=10 images to get the statistical power to support our statement in the manuscript (Fig 3c).

During the revision we have tried to use cholesterol treatment but surprisingly found that it enhances the levels of cGAMP inside the cells (see figure R1 above– not included in the manuscript). We believe this may be due to increased extracellular uptake/delivery of cGAMP to cells and thus generate a bias in our assay.

Direct binding of cholesterol to STING.

1. Figure legend lacks information about these experiments, such as cGAMP dosage and exposure times. We apologize for the lacking information, which have now been added to the legend.

2. The results of the Fig4a and b are overstated (lines 191 and 192), as the differences found in both ER positive curvature and tubular ER are dependent on M β CD exposure, but independent of cGAMP treatment. We have moderated our statement to reflect the data more appropriately. Please see Line 218-226.

In vivo tumour model:

Figure legends have much more detail. Fig4S legend needs correction: there are no panels e and f. Has been corrected.

Cholesterol gene expression associations.

1. Here, authors relate gene expression in their 15 gene candidates as "cholesterol levels". This is not very accurate, as cholesterol levels were not measured. Moreover, although most of their genes belong to either the cholesterol biosynthetic pathway or positive transcriptional regulation

(SREBP2), their upregulation might not reflect increased cholesterol levels, but pretty much the opposite (SREBP2 expression is triggered when intracellular – ER – cholesterol levels are low).

We appreciate the input from the reviewer and agree that our wording have been unprecise. We have modified this to “The group with high level of genes expression associated with the positive regulation of cholesterol metabolism” and “The group with low level of genes expression associated with the positive regulation of cholesterol metabolism” in Fig 6a.

2. Fig7 data does not demonstrate blocking of innate immune responses, as adaptive immune cells can also express chemokines and cytokines and can respond to type I IFN stimulation.

We appreciate the points, and we have made the correction in the text by replacing "innate immune response" with "interferon-inducible immune response”

I also would request clarification on the following minor issues:

- Acronyms definition (ERGIC – line 60, ALPS – line 459) **corrected**

- Line 70-72 lacks references – work describing dying tumour cells’ DNA uptake by immune infiltrating cells and cGAS-STING activation. **Has been corrected with two very relevant review papers**

- What do authors define as “proper immune cells”? (line 91) – this should be properly defined: infiltration, activation, etc. Also, it would clarify if STING activation is required in the tumour cell, the immune cell or both. **We appreciate the comment and have modified the sentence accordingly**

- Line 116: M β CD is not a cholesterol inhibitor (it does not inhibit cholesterol synthesis or import) but it is a cholesterol depletion agent (as authors correctly state in the results section – line 122). This should be corrected. **This has been corrected**

Reviewer #3 (Remarks to the Author):

This is a very interesting and well-written study from Zhang et al establishing a direct effect of cholesterol on STING regulation via cholesterol binding motifs in the protein and changes in ER structure. Further, they showed the STING activating property of cholesterol depletion could enhance tumor control and survival in a mouse model, and that genes regulating cholesterol correlated with ISG induction in human cancer subjects. This study will be of significant interest to the field and has therapeutic implications. There are no major experimental deficiencies, but the description and presentation of the figures need work to improve clarity.

We welcome the positive feedback from the reviewer and agree that there are sections which we can improve and thereby enhance the clarification of our work. We hope the revised manuscript reflect this.

General:

1) In general, could you please spell out acronyms or full molecule names the first time they are used? For instance, SOAT1 in line 112. Spell out LSS before the sentence explaining it. **We thank the reviewer for pointing this out. We have gone through the manuscript once more and made corrections as appropriately.**

2) In multiple figures, it should be “colocalization”, not “colocation”. **We apologize for this misspelling, which has been corrected throughout the figures now.**

3) In multiple figures, the y-axis should go down to 0. Otherwise, it gives a false impression of the magnitude of effect (2a, 2d, 3a, 4f, 4g) **For Fig 2a and 2d (now 2e), we have now re-scaled the Y-axis to 0. For fig 3a, 4f, 4g, for visual interpretation we had modified the y-axis to make clear that we**

explore small but significant changes. Of course, we do not try to make false impressions, which is why we also have clear statistical analysis on the data.

4) In the image stream analyses, it is difficult to appreciate the point – of course cholesterol localizes with ER staining (2a).

Are the 3 rows representing 3 different cell examples of the same condition or 3 different conditions?

Three cell examples for demonstrating the definition of colocalization.

3a and 4a have the same issues. This should be better described in the figure legends.

We have tried to improve the understanding of the legends corresponding to the figures.

In general, the confocal images provide similar information but are much clearer.

We agree that high resolution of confocal images are often more clear. Following this reviewer's suggestions, we have tried to improve and add more input to the confocal images. This is now also supported by quantifications of data. Please see Fig 2b, c, f and Fig 3c.

5) There are multiple instances of blots missing lanes or with inadequate loading (see below).

We have gone through all western blots and redone those that were addressed below.

6) The methods diagrams are OK but need to be accompanied by better written descriptions in either figure legends or methods. For specifics, see below. We thank the reviewer for highlighting this and we have tried to expand all figure legends to help interpret the data better.

7) Any ideas on how cGAMP might activate SOAT1?

This is an excellent scientific question. Based on our current data, we believe it is possible that cGAMP activates STING, leading to the subsequent enhancement of SOAT1 activity. For a more in-depth and specific understanding of the mechanism, we plan to explore it in future research projects. See also our new fig. 2f

8) Any ideas why the low but not high dose M-beta-CD worked in the tumor models? This surprising finding should at least be mentioned in the discussion. We thank the reviewer for highlighting this. We agree that it is a surprising finding that "less is better than more". We currently do not know why but we can speculate. Many drugs are known to have bell-curved functions, which also have been shown with some STING agonists. It is possible that too much depletion of cholesterol in the TME may counteract various immune cells functions that could otherwise support anti-tumoral activities. It may also be that high cholesterol depletion support too much STING activation in the TME which then shift the balance between anti versus pro-tumorigenic functions. For examples it is known that too much inflammation can drive immunosuppressive mechanisms. We have now added a small section in the discussion regarding this. Please see line 407-414.

More specific:

9) In Figure S1a, there is an increase at 4 and 8 hours, but actually a decrease at 20 hours (line 126). Correct, the decrease at 20hrs is related to the degradation of STING following its activation. This has been well-described in the literature as activated STING complexes are pushed toward autophagosomes as part of a negative feedback loop to shut down signaling.

10) In Figure 1c the AAVS1 control needs more description and should be referenced. We have added a sentence and the two references related to targeting AAVS1 as a safe-habour locus. Please see Line 146-147.

11) The graphs in Figure S1b and S1c showing an effect on STING functional output should be included in the main report. We thank the reviewer for suggesting this, which is a valid point. The

figure 1e-f has now been corrected accordingly.

12) Almost the whole paper (Figs 1-4) is in THP1 and HEK293. It would be helpful to incorporate at least a sample of the primary cell work in the supplemental figures back into the main body of the paper. For many methodological papers the use of cell lines are often preferred as it allows better modifications and repetition under same conditions. CRISPR KO in primary macrophages is very difficult and live-view imagine is also very complicated with done on primary cells. To address the reviewer comment, we have moved some of our data with primary human cells to figure 1g.

13) Does exogenous cholesterol have any impact on the biochemical readouts in Figure 1? We have completed this experiment – please see Figure R1. It is clear that exogenous cholesterol has an unexpected impact on the uptake and accumulation of cGAMP. The mechanism for this we do not know.

14) In Figure 1d it looks like there is a band missing for the 2 ug MβCD. Is there another blot you could use? We have now re-crop the blot with more accurate images where no bands are missing.

15) The control for Fig S1d-e (IFN-a) should be indicated in the figure legend. We apologize for the missing information. This is now added in the legend

16) In Figure S2a the figure and legend suggest the MβCD was used alone rather than in combination as suggested on line 141. We thank the reviewer for pointing this out. The legend has been corrected to emphasize that cGAMP was added to all conditions to demonstrate how Filipin-III or MβCD resulted in synergistic effects to cGAMP-mediated STING activation.

17) What happened to the OSBP CRISPR lane in 2c? Is there another blot you could use? We have generated new KO cell lines with two sgRNAs and reconstructed the western blot to replace the old figure. Please see the new Fig 2d.

If these are bulk transfections, the amount of knockout should be shown at least in supplemental data.

We have included a new supplementary figure with ICE analysis to measure the KO efficiency. Please see Fig S2a.

18) In Figure 3b, what is M and UT? This should go in the figure legend. It doesn't look like there is any difference upon adding MβCD. The vesicle budding study is difficult to understand and does not add very much to the report. It needs to be better described or removed.

'M' denotes that only membrane fractions were included in the in vitro reaction, with no cytosol. 'UT' signifies no MβCD treatment. We sincerely apologize for any confusion and have updated the labeling of Fig. 3b. Additionally, we have added specific experimental details to the Figure legend. Furthermore, in response to the issue of unclear differences in blot results, we have optimized and updated this experiment. Please refer to the revised version of Fig3b.

19) In Figure 3c, could you add a STING/PDI column?

Yes, we have included the STING/PDI column as well as statistical analysis of multiple images. Please see Fig 3c.

20) The ALPS-GFP assay requires description. For the tubular ER study (fig. 4b) Did you use anti-RTN as indicated in the figure legend or Nogo, as written on the Figure? How exactly were these “counted” and the tubular to flat ratios derived?

We understand the need for better description. This is now included in both the result section and figure legend as well as in the M&M method. We apologize for any confusion related to the wording of RTN/Nogo. In fact, RTN and Nogo are two names for the same protein, To clarify, we have replaced all 'RTN' with 'Nogo'. The method of counting has been included in the M&M method.

21) The labeling of Figure 4c western blot is not clear. Is this the cholesterol bead ip? And blot with FLAG or with STING? Or a STING ip? Why is the first lane of input missing?

We agree with the reviewer's comment that the labeling in Fig4c is unclear. For the experiment, we first purified and concentrated Flag-STING protein using Flag beads, followed by a pulldown assay with purified protein and cholesterol-coated beads. To address this concern, we have now provided more detailed labeling information for the blots and further improved the description of this experiment in the Materials and Methods section.

22) The generation of the STING mutants should be described. It's missing. We have added a new section in Material and Method describing this process.

23) The individual spaghetti plots do not really add anything to 5b. We included these data as often reviewers ask to see the pattern for individual mice and not just the mean value. However, based on the reviewer comment we have moved these results into a new Fig S7.

24) Where is the results description of 5f? It goes from 5e to 5g. The upper row of 5f is also not good quality. We thank the reviewer for highlighting this, which was a clear mistake as the section had not been merged into our final submitted manuscript. This has now been reintroduced. Please see Line 282-286. Furthermore, we have re-done and updated the top row of Fig. 5f to improve the quality of our images.

25) In 5h, is the blue line superimposed on the red for most of it? We have now dotted the red line to make it clearer that the lines are overlapping. See new fig. 5h.

26) Do you have the Ns for Figure 6a? Please see the number of patients on the right side of the labels.

27) Please keep the color coding consistent between the 6c diagram and 6d and e graphs. It's exactly flipped. This was a clear mistake and has now been corrected.

28) In the human subjects, is there any information on plasma cholesterol levels or statin use? Sadly not as we achieved these results from public databases.

Minor:

1) Line 65: Rather than IFN-I, it would be better to write "type I IFN".

We accept to change the wording

2) Line 94: bell "shaped" rather than bell curved. corrected

3) Line 123: missing the word "using" between cholesterol and Methyl-beta-cyclodextrin. corrected

4) Line 166 should be "exits" rather than "exiting". Sentence reworded

5) Line 231: "promote" rather than "prone"? corrected

6) Line 295: the CRAC motif in the connecting loop (in addition to transmembrane region) should also be mentioned. We are uncertain what the reviewer refers to as the line 295 do not mention anything about the CRAC motif.

REVIEWER COMMENTS

Reviewer #1 (Remarks to the Author):

No more question.

Reviewer #2 (Remarks to the Author):

After this extensive review, the authors have addressed the majority of the comments raised and the manuscript is better placed for publication. I still think that some of the conclusions are overstated and not supported by the data presented. In particular, because cholesterol content has not been quantified, I still think that the authors cannot conclude that this is a specific effect of cholesterol depleted in the ER, neither can authors state that filipin III depletes cholesterol (only that its biological effect overlaps with MbCD on certain phenotypes tested). There are still some graphs where y axis is not 0 (I understand the authors' comments, but I still think that the representation should be consistent overall) and the manuscript still contains several typos.

Reviewer #3 (Remarks to the Author):

I would like to thank the authors for their attention to all of the requests. New confocal images are very helpful and figure clarity much improved throughout. The findings supporting regulation of STING signaling by cholesterol levels are sufficiently compelling, and some players identified (e.g. SOAT1 and motifs in STING). The capacity for synergism in the tumor model is particularly intriguing. Yes, work needs to be done to clarify these mechanisms, and reasons behind odd dose response, but would agree it's beyond the current scope.

Just a few minor details:

- 1) Y-axis needs to go to 0 in Figs 3a and 4g. It's deceptive otherwise. The bars and stars will attest to statistical significance.
- 2) In Fig.1, the letters "ti" have become a box with an x through it. Hopefully that's just an artifact in my pdf.
- 3) Colocalization misspelled in 3a (missing an a) and still "colocation" in 4g and h.
- 4) In 1a, the 0 and 0.5 run together looking like an 8.

REVIEWER COMMENTS

Reviewer #1 (Remarks to the Author):

No more question.

Reviewer #2 (Remarks to the Author):

After this extensive review, the authors have addressed the majority of the comments raised and the manuscript is better placed for publication. I still think that some of the conclusions are over-stated and not supported by the data presented. In particular, because cholesterol content has not been quantified, I still think that the authors cannot conclude that this is a specific effect of cholesterol depleted in the ER, neither can authors state that filipin III depletes cholesterol (only that its biological effect overlaps with MbCD on certain phenotypes tested).

We highly appreciate the comment made by reviewer 2 regarding the lack of quantitative measurement of ER cholesterol levels. We believe that directly measuring the absolute cholesterol content in the ER is unfeasible. Firstly, the proportion of cholesterol in the endoplasmic reticulum (ER) constitutes only about 0.5% of the entire cell's cholesterol content. Secondly, we are not aware of any protocol available that will allow us to isolate the ER in an absolutely pure form and thereby measure cholesterol. Given the inherently low cholesterol content in the ER coupled with the inability to achieve absolute purity of the ER, attempting direct measurement of cholesterol would introduce significant errors. Therefore, we argue that the indirect quantitative methods we are employing, such as ImageStream and Confocal microscopy offers a relatively accurate means of quantifying ER cholesterol.

We have softened the language related to the conclusions based on our data analysis. Please see line 153-156, line 178-179, and line 183.

There are still some graphs where y axis is not 0 (I understand the authors' comments, but I still think that the representation should be consistent overall) and the manuscript still contains several typos.

Following the suggestion of reviewer 2, we have adjusted all graphs to have a y-axis starting from 0.

Reviewer #3 (Remarks to the Author):

I would like to thank the authors for their attention to all of the requests. New confocal images are very helpful and figure clarity much improved throughout. The findings supporting regulation of STING signaling by cholesterol levels are sufficiently compelling, and some players identified (e.g. SOAT1 and motifs in STING). The capacity for synergism in the tumor model is particularly intriguing. Yes, work needs to be done to clarify these mechanisms, and reasons behind odd dose response, but would agree it's beyond the current scope.

Just a few minor details:

1) Y-axis needs to go to 0 in Figs 3a and 4g. It's deceptive otherwise. The bars and stars will attest to statistical significance.

We would like to thank reviewer 3 for the support of our manuscript. As suggested, we have now adjusted the y-axes of Fig 3a and 4g to start from zero. Previously, our intention was solely to make the differences in data more visually apparent and clear, without any 'deceptive' intent.

2) In Fig.1, the letters "ti" have become a box with an x through it. Hopefully that's just an artifact in my pdf.

We double-checked Fig 1 and found that the 'ti' is displaying normally.

3) Colocalization misspelled in 3a (missing an a) and still "colocation" in 4g and h.

Thank you very much for pointing out these errors. These are now corrected.

4) In 1a, the 0 and 0.5 run together looking like an 8.

We have improved the presentation of the x-axis numbers.

REVIEWERS' COMMENTS

Reviewer #2 (Remarks to the Author):

The authorst have correclty addressed all my comments.

Reviewer #3 (Remarks to the Author):

Thank you for making the changes. No more concerns.